# Turning the tide: comparison of tidal flow by periodic sealevel fluctuation and by periodic bed tilting in scaled landscape experiments of estuaries

Maarten G. Kleinhans[1], Maarten van der Vegt[1], Jasper Leuven[1], Lisanne Braat[1], Henk Markies[1], Arjan Simmelink[2], Chris Roosendaal[1], Arjan van Eijk[1], Paul Vrijbergen[1], and Marcel van Maarseveen[1]

[1]Faculty of Geosciences, Utrecht University, PObox 80115, 3508 TC Utrecht, The Netherlands
[2]formerly at Consmema design and steel construction, Hattem, The Netherlands

*Correspondence to:* M.G. Kleinhans, m.g.kleinhans@uu.nl

**Abstract.** Analogue models or scale experiments of estuaries and short tidal basins are notoriously difficult to create in the laboratory because of the difficulty to obtain currents strong enough to transport sand. Our recently discovered method to drive tidal currents by periodically tilting the entire flume leads to intense sediment transport in both the ebb and flood phase causing dynamic channel and shoal patterns. However, it remains unclear whether tilting produces periodic flows with characteristic tidal properties that are sufficiently similar to those in nature for the purpose of landscape experiments. Moreover, it is not well understood why the flows driven by periodic sealevel fluctuation, as in nature, are not sufficient for morphodynamic experiments. Here we compare for the first time the tidal currents driven by sealevel fluctuations and by tilting. Experiments were run in a 20 by 3 m straight flume, the Metronome, for a range of tilting periods and one or two boundaries open at constant head with free inflow and outflow. Also experiments were run with flow driven by periodic sealevel fluctuations. We recorded surface flow velocity along the flume with Particle Imaging Velocimetry and measured water levels along the flume. We compared the results to a one-dimensional model with shallow flow equations for a rough bed, which was tested on the experiments and applied to a range of length scales bridging small experiments and large estuaries. We found that Reynolds' method results in negligible flows along the flume except for the first few meters, whereas flume tilting results in nearly uniform, reversing flow velocities along the entire flume that are strong enough to move sand. Furthermore, tidal excursion length relative to basin length and the dominance of friction over inertia is similar in tidal experiments and reality. The sediment mobility converges between Reynolds' method and tilting for flumes of hundreds of meters long, which is impractical. Smaller flumes of a few meters length on the other hand are much more dominated by friction than natural systems, meaning that sediment suspension would be impossible in the resulting laminar flow on tidal flats. Where Reynolds' method is limited by small sediment mobility and high tidal range relative to water depth, the tilting method allows independent control over the variables flow depth, velocity, sediment mobility, tidal period and excursion length, and tidal asymmetry. A periodically tilting flume thus opens up the possibility of systematic biogeomorphological experimentation with self-formed estuaries.

# 1 Introduction

## 1.1 Problem definition

Estuaries are tidal basins with some freshwater inflow, that are long relative to their inlet width. Reversing tidal flow is driven by the tidal water level fluctuations at the seaward boundary. In nature, estuaries have embayed or seaward widening planforms
with coastal inlets and are partly filled with intricate patterns of shoals, tidal sand bars, mud flats and tidal marshes. The large-scale planform shape and bar-channel patterns within evolved by biogeomorphological processes and are partly determined by inherited initial conditions and changing boundary conditions (Townend, 2012; de Haas et al., 2017). Certain phenomena are unique to estuaries, such as mutually evasive ebb- or flood-dominated channels separated by shoals (van Veen, 1950; Leuven et al., 2016). These shoals hinder shipping and at the same time may be important habitats. However, gaining understanding
of their behaviour is challenging because modelling sediment transport processes in three-dimensional reversing flow remains overly sensitive to sediment transport parameters (van der Wegen and Roelvink, 2012), and field observations of morphological development spanning decades to centuries are unavailable (Wang et al., 2002; Swinkels et al., 2009). The third, complementary method of research is controlled laboratory experiments (Paola et al., 2009; Stefanon et al., 2010; Kleinhans et al., 2010, 2015a), which are rare for estuarine phenomena in contrast to the large number of river experiments.

Only two sets of experiments simulating estuarine morphodynamics are accessible in the literature: Reynolds (1887, 1889) conducted a large number of experiments in basins of various shapes, and Tambroni et al. (2005) conducted two experiments in an exponentially widening estuary. In both cases the flow was driven by periodic sealevel fluctuations but sediment mobility was too low compared to natural systems. Recently, an alternative experimental method was discovered (Kleinhans et al., 2012, 2014b, 2015b) that caused sufficiently strong reversing flow for sediment transport similarity in tidal inlets by tilting the
entire flume periodically. With relatively small setups this was shown to result in dynamic channel and shoal patterns that are similar to those in tidal inlet systems in nature. However, the experimental tilting principle was not yet applied to estuaries that are much longer than tidal inlets and therefore require much better control on the tidal wave dynamics than short basins (Friedrichs, 2010). Gentle tilting drives the flow in a fundamentally different way than tides do in nature, raising the question to what degree this method leads to similar spatial flow and sediment motion patterns. Moreover, why Reynolds' method does
not lead to sufficient sediment mobility remained unresolved for over a century. Here we compare for the first time the flow in idealised experimental estuaries measured in both the classic Reynolds' setup and the novel tilting setup and extend the comparison by modelling from the smallest experiments to large estuaries.

## 1.2 Targets for tidal landscape experiments

Tidal flows in natural estuaries can be complex but for the purpose of pioneering laboratory experiments we focus firstly on
the most fundamental properties. In nature, flow is mostly driven by a primary tide causing periodic sealevel fluctuation, which propagates as a wave through the estuary mouth. This is modified by other tidal components, river flow and by circulation in deep estuaries with salinity stratification (Dronkers, 1986; Friedrichs and Aubrey, 1988; Friedrichs, 2010; Savenije, 2015). The length $L$ of an estuary is typically up to half a tidal wavelength $L_t$, which is estimated as $L_t = T\sqrt{gh}$ with primary tidal

period $T$ and shallow water wave celerity $\sqrt{gh}$ where $h$ is water depth and $g$ is gravitational acceleration. The tidal amplitude $a$ is usually less than half the water depth (Friedrichs, 2010; Savenije, 2015). The resulting flow velocity $u$ depends on tidal period and cross-sectional area $A$ of the inlet, and on the tidal prism that depends on the planform geometry of the estuary (e.g. Townend, 2012). Typically estuaries get narrower and shallower in landward direction as freely erodible substrates adapt

to spatial gradients in flow velocity, so that flow velocity in many estuaries does not vary more than an order of magnitude with distance from the inlet (Savenije, 2015). The aerial extent and elevation of channels, shoals, mud flats and salt marshes modifies the magnitude, timing and duration of the ebb-directed flow and the flood-directed flow, particularly if these vary along the estuary and if the channels are dredged (Wang et al., 2002; Friedrichs, 2010).

     In turn, sediment mobility and transport are driven by the flow to cause morphological change. Here, mobility is expressed

as the Shields number $\theta = \tau/g(\rho_s - \rho)D$, where $\tau$ is the bed shear stress by the flow, $g =$9.8 m/s$^2$ is gravitational acceleration, $\rho$ and $\rho_s$ are the density of water and sediment, respectively, and $D$ is a representative particle size of the bed sediment. The critical Shields number for the onset of sediment motion is about $\theta_c \approx 0.04$. The bed shear stress is calculated as $\tau = \rho f u|u|$ or $\tau = \rho g u|u|/C^2$ where $u$ is depth-averaged flow velocity driven by the energy gradient, $f$ is a dimensionless bottom drag coefficient and $C$ is the Chézy coefficient with $f = g/C^2$ by definition. The characteristic timescale of large-scale morpho-

logical change is much larger than the tidal period (van der Wegen et al., 2008; Lanzoni and Seminara, 2002). This has an important consequence for modelling and experimentation: the time-dependent phase differences of flow velocity as a function of distance from the mouth are not of first-order importance for the morphodynamics, as long as spatial variations in veloc- ity, residual currents and the resulting sediment mobility are present. This conclusion can also be drawn from linear theory for tidal bar properties, for which a rigid lid flow assumption was sufficient, meaning that water surface fluctuations are only

of secondary importance (Schramkowski et al., 2002; Leuven et al., 2016). The reversing tidal flow is therefore much more important for morphology than the tidal wave behaviour.

     The prime challenge for morphodynamic tidal experiments is that the reversing flow should cause sufficient bed shear stress for periodically reversing sediment motion (Kleinhans et al., 2015b). As the bed sediment calibre cannot be scaled with the same ratio $n_L$ as the dimensions of the system, the energy gradient $S$ of the laboratory system must be increased such that

the mobility $\theta$ remains the same as in the prototype. For medium sands this slope typically is $S = 0.01$ m/m accounting for both particle weight and the large bed roughness in experiments (Kleinhans et al., 2014a). This required energy slope for mobile sediment is well feasible in river experiments, but not in estuary experiments driven by periodic sealevel fluctuation for the following reasons (Fig. 1). Consider an experimental tidal system with a depth of $h =$2 cm. Here the tidal water level amplitude can be at most about $a =$1 cm meaning that $a/h$ is as large as 0.5. Given a typical aspect ratio of the estuary mouth

of $W/h > 100$, this means that the width of the experiment should be about $W =$1 m. However, with a minimum slope of 0.01 m/m, the distance from the mouth with sufficient gradient to move sediment is effectively only about 1 m given the maximum water surface amplitude. In laboratory-sized systems this creates a short tidal basin (Stefanon et al., 2010; Kleinhans et al., 2015b) rather than the long estuary we aim for. This, in turn, leads to a number of other scale problems. The first is a problem with tidal period. For the 2 cm depth and 1 m basin length the required tidal period is about 4-9 s, which is very fast

for water to accelerate and for pumps to deal with. The second problem is that this wave causes very low flow velocities of

$O(10^{-3})$ m/s, which is far below that required for sediment motion. When the tidal amplitude is enlarged with $a/h > 0.5$, a new problem arises: the flow causes net export of sediment results on the seaward sloping bed so that the tidal system excavates until it is in static equilibrium as probably happened in a number of the experiments of Reynolds (1889).

An expensive solution would perhaps be to make impractical experimental setups of $O(10^2)$ m long, which renders morphological time scales impractically long and requires very large pumping capacity. The tilting flume principle, on the other hand, has been shown qualitatively to attain the required sediment mobility, but the principle is counterintuitive: the real world does not tilt periodically. This may be the reason that this principle was not invented in the past 130 years. The fundamentally different driving mechanism for the flow raises the question to what degree the tidal flow is similar to that in nature, and what consequences this may have for the morphological development of estuaries at the experimental scale.

## 1.3  Objectives and approach

In our preliminary work in small flumes so far a comparison between periodic sealevel and periodic tilting was done only qualitatively. Measurements are needed of flow velocity fields and flow depth for a more in-depth analysis of the laboratory flow behavior compared to natural tidal systems, and a larger facility with a higher tilting frequency is needed for better scaling of basin size relative to tidal wavelength. Furthermore, a numerical model reproducing the main dynamics of the experiments is needed to assess whether the tidal flows in the Metronome are similar to those in nature, and, if so, to scale up from the smallest laboratory experiments to the largest natural tidal systems on the planet in order to uncover possible scale problems.

The first objective of this paper is to compare tidal flows generated in the tilting flume and in the Reynolds flume, focussing on the magnitude of reversing flow velocity and sediment mobility along the estuary. To this end we present flow measurements in a large tilting flume facility. Specifically, experiments were designed to directly compare tidal wave behaviour, flow velocity magnitude and tidal asymmetry driven by periodic tilting or sealevel fluctuation in idealised straight tidal channels with rough beds and the largest possible tidal amplitudes. To exclude complex morphodynamic feedbacks, this study is limited to idealised channels without bars and shoals and with fixed rough beds or natural sand beds with conditions below the beginning of sediment motion. The second objective is to assess how the dimensions and dynamics of experiments are to be scaled up to prototype systems, and at what cost in terms of scale problems and distortions. To this end we adapted a one-dimensional model of the shallow water equations to include bed tilting and verified whether the most important tidal behaviour is reproduced. We then compared modelled flow in systems with length scales ranging from a small laboratory setup to a large natural estuary and with flow driven by both methods, characterised by morphologically relevant dimensionless variables.

## 2  Design of the Metronome Facility

Between 2014–2015 we constructed the Metronome, a 20 m long flume designed for periodic tilting to create tidal systems (http://www.uu.nl/metronome, Fig. 2, building plans in online supplements). The basic components are a steel basin that tilts over the short central axis, motion control, water recirculation with a constant head condition at the seaward end, and optical imaging (Fig. 3).

The principle can be reproduced by simple means, namely by any basin, stream table or flume that can be tilted over its axis, a consumer-grade garden pond pump and a camera. The periodic tilting can be driven by an actuator, an excenter mechanism or an adjustable stroke mechanism with a gearbox and motor to drive sinusoidal motion with a period of tens of seconds and a slope amplitude up to 0.02. For the purposes of future reference for ongoing biomorphological experiments and for replication
in other laboratories the specific design of the Metronome is briefly described below.

The steel basin has inner dimensions of 20.00 m long by 3.00 m wide and 0.40 m deep. The flume has two end tanks for water supply, water level control and outflow and for sediment trapping. The tilting axis is directly below the steel floor to minimise longitudinal motion and the entire flume setup is symmetrical about this axis. The basin was constructed from 4 mm stainless steel plates cut and folded such that the sidewalls are suitable for a gantry to screed the bed and set up measurement equipment,
and are a structural part of the basin to minimise bending. Further stiffness was accomplished by a ribbed structure and steel beams along and across the flume. Finite element modelling on the design showed that the maximum expected bending of the flume was at most 2 mm under extreme loads in emergency conditions. This model was also used to select the required range and power of the actuators and motion control and to estimate the loading and required reinforcement of the floor. The steel basin was curved slightly upwards during production such that it is straight under typical water and sediment loading when
supported by the tilting axis.

The end tanks were designed to function as constant head tanks, with sediment traps at the inside of a movable weirs. Water is supplied by four garden pond pumps with a maximum discharge of 4.7 L/s or 9 m head each in a 12 m$^3$ sump tank, which is an inflatable swimming pool in the basement of the building. The 3 m wide weirs in both end tanks are broad-crested with a length of 0.06 m and rounded edges due to the folding of the steel plate from which it was constructed. Small actuators control
the motion of the weirs. This setup means that the flow is critical on the broad-crested weirs, so that the water depth $h_c$ at the weir depends on the specific discharge $q = uh$ over it as $h_c = \left(q^2/g\right)^{1/3}$. Consequently, the water surface elevation at the seaward boundary is not exactly as set by the weir height but is modified slightly depending on the tidal prism. The effect of this will be taken into account in the interpretation of the results herein. In future live-bed experiments the water depth can be corrected by motion of the weir at an appropriate amplitude and phase shift relative to the tilting motion depending on the tidal
prism, such that the water level at the shoreline of the live bed remains approximately constant.

The four actuators to tilt the flume operate in pairs with motion mirrored at the tilting axis. The maximum force is 20 kN, but in downward direction had to be limited as reinforced concrete floor supports the downward force well but the upward, pulling force, not very well. The motion and forces are monitored and internal safety controls prevent values above this that might be damaging. The motion at periods and amplitudes as used in this paper is typically 0.01 mm accurate. The actuators
keep repeatable positions at all times, also during rest, such that the flume does not deform. We found that the flume was best set horizontal through manual measurement with a leveller and 0.5 mm graded rulers on the sand screed riding on the cart, and applying offset positions to all four tilting actuators.

Landscape experiments often show channels clinging to sidewalls, and, when insufficient sand is used, channels that erode down to the flume floor where erosion is enhanced because of the smooth surface. Using groynes or ribs are not solving this
because these force their own patterns on the flow and morphology. A surface of uniform roughness is needed with a roughness

scale larger than the viscous sublayer thickness, much smaller than the smallest bedforms (Kleinhans et al., 2017), and with a gradual transition from an alluviated sand-bed to a fixed rough surface. We therefore covered the Metronome floor and sidewalls with small-scale uniform roughness: an artificial grass of about $15 \pm 1$ mm ($1\sigma$) high, spatially uniform in stem density, glued to the floor in places and further kept down by a few mm of sand. The grass is so stiff that it did not bend noticeably in the strongest experimental flows tested. The glue was applied such that water cannot flow under the grass. An alternative would be sandpaper or any other rough surface but a practical advantage of the artificial grass is easier sand removal and flume floor protection against shovels. We used the grass roughness in the fixed-bed experiments and buried it under sand in the sand-bed experiments. The sand has a $D_{10}$ of 0.33 mm, a $D_{50}$ of 0.57 mm and a $D_{90}$ of 1.2 mm, which has a larger roughness length than the viscous sublayer. We will assume the same sediment properties for all modelled estuaries independent of length scale.

## 3   Experimental Setup and Materials

### 3.1   Geometry and flow conditions

We conducted experiments with various initial and boundary conditions (Table 1). Most importantly, we applied periodic tilting and periodic sealevel fluctuation for comparison. Both experimental approaches were applied on a sand-bed and on a rough, artificial grass bed. The majority of experiments were conducted on the artificial grass bed because this allowed most freedom in conditions that would have led to significant sediment motion on the sand bed. We tested two different boundary conditions for the tilting experiments with grass bed: one open sea boundary and one reflective boundary to represent an estuary with the landward boundary closed, and two open boundaries to represent a reach within a long estuary. We conducted auxiliary experiments with a constant flume gradient to test the flow resistance formulation for the artificial grass bed. The sand-bed experiments were conducted to assess effects of the typical roughness in live-bed experiments on the flow and had one open boundary in all cases. The sand-bed was pre-soaked.

The precise geometry of an estuary strongly determines tidal flow patterns along the river. Given the aim in this paper, we chose the simplest geometries and boundary conditions possible: straight channels and periodic motion (Table 1). An alternative could have been to create exponentially convergent estuaries where the friction loss in the tidal wave is compensated by the landward narrowing such that the flow velocity amplitude is about constant along the estuary (Savenije, 2015). However, this requires careful matching of the convergence length with the tidal conditions and effective friction, which we did not know in advance. Moreover, we do not know yet whether this shape is applicable in the tilting flume. For the artificial grass bed experiments, a straight channel of 0.7 m wide was sectioned off from the remainder of the flume by 0.1 m diameter cotton hoses filled with sand pressed down into the grass. For the sand bed experiments, a channel of 0.6 m wide and 0.03 m deep was carved in a 0.065 m thick sand-bed over the first 18 m of the flume, leaving a basin of 2 m long and 3 m wide uncovered. This 'sea' allows ebb delta formation in future live-bed experiments. The side effect is that the mass of water available for inflow and outflow of the channel is unhindered by the limited capacity of the pumps and the critical flow condition on the weir, making this setup insensitive to adverse seaward boundary effects.

The average water depths were set at about $h =$0.028 m in all grass experiments by applying the same (average) heights of the weirs to submerge the vegetation-like roughness at all times. The Reynolds-type experiments were done with a period of 30 s and a seawater surface amplitude of 0.02 m, that is, the same period as in many of our other experiments and an extreme tidal amplitude, and a less energetic condition with a 60 s period and a 0.01 m amplitude that is closer to conditions in experiments reported in literature. The most basic tilting experiment has two open boundaries with constant elevations of both overflow weirs, approximating constant head conditions. For this condition two experiments were run with tilting periods of 30 s, which is similar to other experiments in literature, and 15 s to investigate the possibility of reducing the tidal excursion length estimated as $L_e \approx uT/2$. In most grass-bed experiments a tilting slope amplitude (maximum slope during sinusoidal tilting) of $9.1 \times 10^{-2}$ m/m was applied. The second setup has one boundary closed (here at 0 m) and the other open, representing a tidal basin of finite length. Here again 15 and 30 s periods were applied. As a control experiment to test the friction relation, the steady flow was measured at constant slopes of $0.91 \times 10^{-3}$ m/m and $2.3 \times 10^{-3}$ m/m and the same water depth as the other experiments.

Conditions in the sand-bed experiment were set such that the channel did not overflow and the sediment hardly moved, which was attained at a mean water depth of $h =$0.018 m and a 40 s period, the typical period for live-bed experiments to be reported in future papers. The Reynolds experiment on sand was conducted with $3.5 \times 10^{-3}$ m water surface amplitude to prevent sediment motion. The tilting experiment was conducted with a tilting slope amplitude of $3.6 \times 10^{-2}$ m/m, for which we observed no significant sediment motion.

## 3.2  Imaging, measurements and data reduction

Flow was measured by water depth measurements and large-scale surface Particle Imaging Velocimetry (PIV) (by the same method as Blanckaert et al., 2012; Marra et al., 2014). The PIV was conducted by spreading white floating particles on the water surface of the flume, repeated photography and image processing to obtain the motion of the particles as detailed below.

Water depth was measured in the grass-bed experiments at various locations along the flume with rulers with 0.5 mm grading supported by small thin-legged tripods. This is rather inaccurate because of irregularities in bed elevation and because the meniscus of the water surface on the rulers. These data were detrended with still water measurements. In the sand-bed experiments conducted later we measured water surface elevation relative to still water with an ultrasonic device at a sound frequency 150 kHz mounted on the tilting flume. The distance of about 0.2 m from the bed with temperature-corrected distance measurement resulted in a footprint of about 0.03 m and a vertical accuracy of about 1 mm. Measurements were collected for three tidal cycles at 10 Hz sampling frequency in phase with the tilting and phase-averaged by fitting with a spline at 1 s interval for presentation.

For the PIV, seven industrial cameras were mounted 3.7 m above the floor of the flume, approximately above the centerline at equal distances. However, camera alignment was hampered by the roof supports in the temporary lab location so that axis positions and directions differ between cameras and are not perpendicular to the flume floor. This caused the geometry of the optical system relative to the flume to be suboptimal, resulting for higher tilt angles in a few pixels mismatch between adjacent cameras. This does not affect the conclusions of this paper because the velocity is spatially averaged along the flume and the

focus is on general characteristics and behaviour. The cameras are CMOS MAKO color cameras with a resolution of 2048 by 2048 pixels. The cameras have a lens with fixed focal length of 12.5 mm. The footprint is about 3.15 m, so that a pixel on average covers about 1.5–2 mm. Hardware and software are designed to allow simultaneous 25 Hz imaging for the purpose of PIV. The trigger for the cameras is taken from the tilting motor controller at exactly defined moments in the tidal cycle. For the PIV this trigger starts a 25 Hz pulse train from a frequency generator in order to have accurate, computer clock-independent timing.

The flume is illuminated at about 600 lux with daylight-coloured TL aimed at a white diffusive ceiling at about 4.5 m above the flume floor, designed for imaging as well as future vegetation growth. This allows for low exposure durations, but we later found that the ceiling reflected on the water surface to hinder imaging in live-bed experiments. By the time we conducted the additional sand-bed experiments a diffusive white sheet was suspended below the ceiling and lamps in the shape of a tent. This improved lighting although it reduced light intensity, but this did not affect the PIV imaging.

The procedure for data collection was as follows. White floating particles of 2–3 mm diameter were seeded on the water surface along the flume, and newly supplied by operators at both boundaries where necessary. After about five tidal cycles the flow was considered in equilibrium. In 16 phases of the tide, ten images were collected at 25 Hz simultaneously by all cameras. Water levels were measured before seeding the PIV particles. Control experiments with constant slope were conducted in the same manner but with lower slopes because of the rapid evacuation of floating particles.

Raw images were first debayered to obtain RGB color images, from which only the green layer was taken for analysis. Background images were subtracted that were obtained for the same tidal phase without floating particles. These images were then rectified using the Caltech camera calibration toolbox in Matlab (http://www.vision.caltech.edu/bouguetj/calib_doc/, version October 15, 2004), after obtaining camera calibrations.

Flow velocities were calculated for every pair of consecutive images using the MPIV toolbox in Matlab (Mori and Chang (2003), http://www.oceanwave.jp/softwares/mpiv/). The focus of this paper is on width-averaged flow in a uniform channel, so that the conventional cross-correlation algorithm for PIV suffices. This means that the peak cross-correlation is used as mean particle displacement in a given window. This was run with a window size of 100 pixels with 50% overlap. Subsequently the vector fields were scaled by the footprint of the cameras which was calculated from the geometry of the flume, average height of cameras and camera resolution and the instantaneous tilting angle. As a result flow velocity vectors were determined on a regular grid at about 77 mm spacing. Erroneous vectors resulted from windows that were partially filled with flume wall, spots empty of particles, mismatched particles and reflections on the water surface. After filtering out the 1% most extreme values that are assumed to be errors, width-averaged velocities were obtained along the flume for each cross-section within 0.36 s at 16 phases in the tidal cycle.

## 4 Numerical Flow Model

### 4.1 Model formulation

We use a one-dimensional model that has been demonstrated to reproduce the most important tidal dynamics (Friedrichs and Aubrey, 1988). We assume a rectangular channel of constant width and depth and solve the shallow water equations for friction-dominated conditions; a condition that we will check later. Here we modify the model to tilt the bed periodically so that it can be applied to the tilting flume.

Continuity is conserved as:

$$w\frac{\partial \eta + z_b}{\partial t} = \frac{\partial uw(\eta + z_b)}{\partial x} \tag{1}$$

where $w$ =width, $h$ =depth, $u$ =flow velocity, water depth $h = \eta + z_b$, with water surface located at level $z = \eta$, bed level at $z = -z_b$, $t$ = time and $x$ = streamwise coordinate. Here, cross-sectional area $A = hW$ and discharge $Q = uhW$ in our rectangular channel. The left-hand side represents the time rate of change of the wetted cross-sectional area, given constant width entirely due to water level changes, and the right-hand side represents volume flux convergence along the channel. To tilt the system periodically, $z_b$ is imposed as a function of time and therefore changes at the tidal time scale, in contrast to most studies where it evolves on a much longer morphodynamic time scale.

The momentum balance equation is given as:

$$\frac{\partial Q}{\partial t} + gA\frac{\partial \eta}{\partial x} + g\frac{Q|Q|P}{C^2 A^2} = 0 \tag{2}$$

where the terms from left to right represent local acceleration, along-channel pressure gradient and bottom friction. Furthermore, $C$ = Chézy roughness coefficient and $P = 2h + W$ is the wetted perimeter for the rectangular cross-section.

We excluded advection because this term is an order of magnitude smaller than the inertia, friction and pressure gradient terms. Inertia will scale as [U]/[T], where [U] is the typical velocity scale and [T] is the time scale over which the velocity changes, typically a quarter of the tilting period. Advection scales as [U$^2$]/[L$_x$], where L$_x$ is a typical length scale. For the Metronome that will be in the order of half the length of the basin. Hence, advection with respect to inertia scales as [UT]/[2L$_x$], which is half the tidal excursion length divided by the basin length. For typical conditions in the Metronome this ratio is smaller than 0.1–0.2. In order to keep this scale the same in smaller flumes, the tilting period needs to be reduced linearly with the flume length.

The set of equations are discretised on a staggered grid with $n$ flux points and $n - 1$ bed elevation points and solved by an explicit numerical scheme that is second order in both time and space. The condition that Courant numbers for surface wave celerity and flow velocity are below unity was checked for every model run. Typical model settings are time step $dt$ =0.05 s and a spatial step $dx$ =0.05 m for a domain of $L = 20$ m with $T = 40$ s and $h = 0.025$ m. We assume a constant width in all cases.

Three sets of boundary conditions simulate three different experimental setups. In Reynold's setup, the bed $z_b$ is static, the landward boundary is closed and the seaward water level is a function of time, i.e. $\eta = a\sin(2\pi t/T)$ at $x = L$. As in nature,

this enforces the pressure gradient at the seaward boundary only, neglecting upstream rivers. The other two setups are tilting basins with one or two boundaries open where $z_b$ is a function of time, for example at $x = 0$ m, $z_b = z_0 + a \sin(2\pi t/T)$, with $z_0$ being the position of the bed at zero tilt. This means that the pressure gradient is enforced to be equal along the flume, and that water depth can change because both bed-level and water surface are function of time and space. Two tilting scenarios applied:

one with the landward boundary closed and the seaward boundary open with a fixed water depth and free flux (abbreviated henceforth as 'tilt1'), and one with both the landward and seaward boundary open ('tilt2').

## 4.2   Hydraulic resistance

The artificial grass cover of the Metronome floor causes hydraulic resistance similar to that of submerged unbending vegetation. This flow resistance is calculated with the relation found by Baptist et al. (2006, their Eq. 74). Furthermore, the surface flow

velocity is derived from the model calculations in order to be able to compare with the PIV data.

The Chézy roughness coefficient for submerged vegetation is calculated as the combined effect of bottom roughness, through-flow resistance and overflow resistance (Fig. 4a):

$$C = \sqrt{\left( \frac{1}{C_b^2} + \frac{c_D N_s D_s H_s}{2g} \right)^{-1}} + \frac{\sqrt{g}}{\kappa} \ln \frac{h}{H_s} \tag{3}$$

where $N_s$ =number of stems, here measured at 50,000 m$^{-2}$, $D_s$ =stem diameter, here measured at 0.4 mm, $H_s$ =vegetation

height, here 14 mm, and $\kappa = 0.4$ is Karman's constant. The first term represents the bed friction below the vegetation; the second term represents the friction for flow through the vegetation and the third term represents friction for flow over the vegetation. The drag coefficient $c_D$ of vegetation is here made dependent on the Reynolds number $Re = uh/\nu$ with $\nu = 1 \times 10^{-6}$, because during flow reversal it may well drop below typical turbulent flow values. The drag coefficient is dynamically calculated with a Coleman-type constitutive relation:

$$c_D = 1 + \frac{30}{Re} + \frac{15}{Re^{0.6}} \tag{4}$$

so that for high $Re$, $c_D \approx 1$. We assume a minimum $Re = 30$ so that the maximum $c_D \approx 4$, which occurs for velocities below about 0.001 m/s. The roughness of the bottom of the vegetated layer is calculated as:

$$C_b = 18_{10} \log \frac{12h}{k_s} \tag{5}$$

where $k_s$ =is the Nikuradse roughness length, here taken to be equal to the 90th percentile of the particle size distribution.

In the sand-bed experiments a constant $C = 25 \sqrt{m}/s$ was assumed. The dimensionless friction factor is calculated from the Chézy coefficient as $f = g/C^2$.

To be of use for the present purpose, the flow velocity at the water surface is needed for comparison of model results with PIV-derived data. Corrections usually reported in literature assume a logarithmic flow velocity profile, but in the present case a layer of water is 'skimming' over the vegetation so that the partitioning of flow between lower and higher layers differs. A

correction factor was therefore calculated by combination of equations in Baptist et al. (2006) for a range of water depths above

the submergence height of the vegetation. The ratio of surface velocity and depth-averaged velocity was found to be insensitive to total water depth (Fig. 4b), meaning that water depth variations during the tidal cycle do not change the ratio between depth-averaged and surface velocity more than, say, 5–10%. Here flows with emergent vegetation were avoided because the present method of PIV is impossible to use under these conditions. Furthermore, the flow is not well described for the transition between barely submerged vegetation to emerged vegetation where the aforementioned ratio rapidly drops to unity, so these conditions are also avoided. In the remainder of this paper the modelled velocities are corrected with a constant multiplication factor of 1.95 for the grass-bed experiments and 1.60 for the sand-bed experiments, which leads to an estimated error smaller than $\pm 5\%$ for the lowest and highest water levels, respectively.

## 5    Experimental results and comparison to model results

Comparison of all experiments shows that flow velocities in the tilting flume are much larger than in the Reynolds setup. High velocities occur nearly simultaneous along the flume as expected because it is driven by the gradient of the entire flume rather than the gradient caused by a tidal wave initiated at the seaward boundary. These results are consistent with the numerical model. The model scenarios of initial conditions and boundary conditions are the same as in the experiments. Below the results are described and compared.

### 5.1    Tilting flume experiments with two open boundaries

The tilting with two open boundaries shows nearly symmetrical reversing flow (Figs 6,7). Spatial patterns in flow velocity along the flume appear consistent between tidal phases and with the unidirectional flow experiments and are caused by camera alignment and irregularities on the flume bed. These are further ignored. The flow velocity lags behind the periodic tilting with about 2-3 seconds in both the 30 s and 15 s period tilting (Fig. 6b and Fig. 7b). Measured water level fluctuates periodically near the boundaries, especially at the 20 m boundary (Fig. 6c). The faster tilting (15 s) experiment has a lower velocity amplitude that also occurs nearly simultaneous along the flume. On the other hand the slower tilting has a higher velocity amplitude in the middle of the flume. The slower tilting also shows more deformation in the velocity signal than the faster tilting (compare Fig. 6b and Fig. 7b).

The model results show a fairly simple periodic flow that is nearly uniform along the flume, with a very minor reduction of flow velocity at the boundaries (Fig. 6a). Likewise, the modelled water levels are nearly static (Fig. 6c). Modelled flow velocities fit the observations fairly well when local accelerations due to bed irregularity and discontinuities due to camera positioning are ignored. However, a wave forms at both boundaries in all tilting experiments that lead to velocity peaks coinciding with water level peaks.

The time lag differs between the model and the flume (Fig. 6b), so that the highest velocities of the tidal cycle are approximately modelled correctly but there is a mismatch between model and observations near the slack. The measured flows begin to decelerate sooner after the peak and accelerate slower after the slack, while the modelled flow has a more rapid reversal of flow.

We compared the amplitudes and phases of tidal components in the observed and modelled velocity signals in the middle of the flume (Fig. 8). For clarity the full tilting period of the flume is called 'principal tide' or 'T1' rather than M2. The comparison shows that the tidal velocity signal is dominated by the tilting period. The 'second overtide' (T3 rather than M6) is about 2% of the velocity amplitude due to friction and the 'first overtide' (T2 rather than M4) is even lower due to the negligible water

level fluctuations. For the latter the deviation between modelled and observed velocity is also the largest but this cannot be considered significant given an uncertainty in the velocity data of a few percent. The phase lag of T3 surprisingly is opposite in the model compared to the observations. However, the phase lags are much smaller for the principal tide. Possible causes are discussed later.

## 5.2   Tilting flume experiments with one open boundary

The observed and modelled flows in experiments with one boundary closed are fairly similar to those with two open boundaries with two major differences (Figs 9,10). First, the flow velocity reduces to zero at the closed boundary over a distance of about 1–2 m for the flood current (towards the closed boundary) in both experiments, and increases to its maximum value over a distance of about 5 m for the ebb current in the 30 s experiment and about 3 m in the 15 s experiment. This asymmetry between ebb and flood currents is caused by the fact that water depth increases during the flood stage and decreases during the ebb

stage.

The second difference with the open boundary experiments is the effect of reflection of the tidal wave on the closed boundary. This leads to water depth and velocity fluctuations close to the boundary (Fig. 9). As a result a water surface wave with a velocity peak travels seaward over an 8–10 m distance whilst the tilting slope peaks and reverses, to dampen out at the peak flood velocity. The primary effect of this wave superimposed on the tilting is a reduction of velocity near the upstream

boundary. In the middle and downstream reaches of the flume, the observed and modelled flow shows negligible differences with the cases of two open boundaries. We visually observed that the wave formed a bore of several millimeters high in the experiments.

The modelled and observed water level amplitudes at the upstream boundary agree fairly well (Fig. 9c). The absolute level differs, but this is meaningless in the experiments because the data was detrended.

The harmonic analyses show that the runs with one boundary closed plot close to the runs with both boundaries open for all tidal components, except for the T2 that is two orders of magnitude smaller than the T1 (Fig. 8). This means that the flow in the middle of the flume is not affected by the upstream boundary being closed, in agreement with the observations made above.

## 5.3   Reynolds-type experiments

Flow in the Reynolds setup with periodic sealevel fluctuations is weak (Figs 11). The 30 s experiment with the amplitude

exceeding half a water depth showed effects of drying and flooding, invalidating this experiment for the present purposes. In general the strongest flows are generated at the sealevel boundary, decaying rapidly towards the closed boundary. The data show that velocity halves within the first 3 m in both experiments. Furthermore, a local minimum velocity occurs in the middle

of the flume and a slight increase in flow velocity at one quarter of the length with opposite phase to that at the mouth. The numerical model roughly reproduces this pattern but predicts higher velocities in the upstream half of the flume than observed.

However, this experiment shows a velocity limitation. Even though a 0.01 m sealevel amplitude was imposed, the observed sealevel amplitude at 0.1 m from the upstream boundary is only half this value. This may be due to the pump capacity limitation at the seaward boundary. For this reason we ran the model with half the design amplitude, which resulted in fairly close correspondence of flow velocity in the most seaward few meters. Furthermore, higher modelled water level amplitudes did not result in equally higher flow velocities because of the nonlinear effects of friction in shallower flow.

## 5.4 Sand-bed experiments

The sand-bed experiments with the tilting and Reynolds setups behave largely the same as the grass-bed experiments (Figs 12,13). The flow velocity amplitude is much larger in the tilting experiment than in the Reynolds experiment, despite the modest tilting slope amplitude. Despite the perfectly symmetrical tilting motion, the ebb and flood phases are asymmetrical: flood velocities occur at higher water levels than the same ebb velocities in the first few meters from the closed boundary despite the perfectly symmetrical tilting motion. On the other hand, the velocity amplitude in the Reynolds experiment decays rapidly in landward direction, but the sealevel amplitude is already 30–40% of the water depth and cannot be increased much.

The sand-bed experiments have a complex geometry with a narrow, shallow channel connected to a wide and deep sea. This leads to two-dimensionality in the flow pattern that the one-dimensional model cannot cover well, such as the high peak in modelled flow velocity at the transition from sea to channel, which is gentler in the experiment due to convergence and divergence in the sea. Also the large spread in the flow velocities at $x = 18.6$ m in Fig. 12b and Fig. 13b is due to the two-dimensional variation. The spatial and temporal patterns in the velocity data are qualitatively similar to the model results with magnitudes of flow velocity within about 20%. However, despite the water available in the sea for rapid inflow and outflow of the channel, and the narrowed flume, the inflow velocity again appeared to be limited. When half the sealevel amplitude was imposed in the model, as in the grass-bed experiments, we obtained a velocity and water level amplitude similar to that in the experiments.

The water surface amplitude and phase reasonably modelled in the Reynolds setup (Fig. 12c) was somewhat overestimated at the landward boundary, and imperfectly predicted in the tilting setup, again particularly at the upstream boundary (Fig. 13c). Possible reasons are irregularities in the sand bed. As in the grass-covered experiments, bores of a few millimeter high form (Fig. 12c and Fig. 13c). A small ebb bore initiates near the upstream boundary and a larger flood bore initiates at the seaward boundary. Furthermore, the measured velocity amplitude reduces faster in landward direction than the modelled velocity in the Reynolds experiments. This is surprising, because with a sea present in the tilting flume we do not expect the flux from the seaward boundary to be limited by the pumps so we did expect the measured flow to resemble the model better.

The sand-bed experiments were designed as the initial condition for live-bed experiments to be done later and are in that sense closer to future morphological experiments than the grass-bed experiments. However, the sudden transition from sea to channel renders the data less straightforward to interpret. Nevertheless the general correspondence in behaviour between the

grass-bed and sand-bed experiments and the model runs shows consistent behaviour of the tilting flume in comparison to the Reynolds setup, which allows general conclusions.

The tilting leads to two unexpected effects that need to be cancelled by periodic motion of the overflow weir. Flow depth over the weir is controlled by the specific discharge given that the Froude number remains constant. This means that compensation is required, approximately in phase with the tilting and depending on discharge to maintain constant sealevel. To have space for the development of an ebb delta, a 'sea' needs to be installed over a length of a few meters as in the pilot experiment in Fig. 2. However, the tilting would cause fast outflow into the sea during ebb, which leads to water level change at the coastline. This can be prevented by opposite-phase correction of the downstream weir to maintain constant sealevel at the coastline rather than at the weir.

## 5.5 Control experiments with constant slope

Measured flow in the constant slope experiments is on average uniform as expected (Fig. 5). However, there are spatial variations up to 20% that are consistent between the two experiments for flow velocity and for water depth. Some of the variation occurs at the transitions between camera images, which can be explained by deviations in camera orientation. However, a larger part is also seen in the water depth measurements including the still water depth and can therefore be attributed to irregularities in the elevation of the artificial grass and the thickness of the sand bed. For example, the increased velocity at 16–18 m coincides with shallower flow and the lowest velocities occur at 6–7 m and 12–13 m.

The highest water depths occur at the upstream boundary (0 m) and the downstream boundary (20 m), perhaps because here the grass was glued to the flume floor and the sand was not spread out as well. Water depth is lowest for the highest flow velocity as expected because only slope was changed.

The predicted flow velocity based on measured average water depth and imposed slope is about correct, which we take as sufficient evidence that the measured artificial vegetation characteristics lead to the correct predicted friction coefficient in the model. The effective Chézy coefficient is about 11 $\sqrt{m}/s$ for these conditions, which is typical for shallow flume experiments with rough bed.

## 5.6 Tidal asymmetry

We used the model to explore tidal asymmetry and magnitudes of overtides (extending the range of models shown in Fig. 8). Tidal asymmetry is often used to indicate sedimentation tendencies. This was here calculated as a function of the tilting slope amplitude. Tidal analysis for a range of tilting slopes shows a straightforward increase of velocity amplitude for the main component with only deviatory behaviour within the first meter of the upstream boundary. The higher harmonics, however, do not increase monotonously with tilting amplitude but the velocity amplitudes are at least an order of magnitude smaller than the principal tide.

The model runs further show that ear the closed landward boundary flood velocities are higher and ebb duration is longer, meaning that the head of the estuary fills rapidly with water and empties slowly. This would lead to sedimentation as expected with principal tide and without river inflow. The inlet has approximately symmetrical tides but is slightly ebb-dominant.

Halfway the flume and in upstream direction the currents are ebb-dominated and the ebb duration is also longer than the flood, which is mostly due to a minor second overtide contribution. The behaviour towards, and at, the upstream boundary is sensitive to the tilting amplitude: above the large gradient of 0.02 m/m (results not shown) the currents become flood-dominant and the flood duration exceeds the ebb duration. However, the strongest responses all occur in the upstream few meters of the flume near the closed boundary. Here the experiments show much smaller water surface fluctuations, perhaps because the upstream boundary is less reflective and more subject to friction than in the model. Furthermore, in a live-bed experiment, sedimentation would rapidly modify the morphology, all of which would reduce these asymmetries.

## 5.7 Effects of bed imperfections on the measured flows

The data indicate that irregularities in the grass and sand bed in the flume affected the tidal flows. We tested this by a model run with depth variation along the flume. The sensitivity of the flow to water depth variations is rather large (Fig. 14): a gradually increased depth with a maximum of 5 mm, less than 20% of the original depth, already causes large and unexpected spatial variations in flow velocity and depth. In particular, the increased depth causes increased ebb velocities at the seaward boundary during some phases, and decreased flood velocities. The flow velocity patterns with modified bed elevation are more nonuniform even in the middle of the flume (Fig. 14a) and resemble those observed in the experiments. This makes it likely that the irregularities of the bed in the experiments caused at least some of the deviations between the model and experimental data, in addition to potential bias in the data due to imperfect camera positioning and calibration.

The model generally reproduces tidal dynamics in the experiments. Two main differences emerged between the model and the data that need to be taken into account in interpretations. Firstly, the water level amplitude is smaller in the experiments than in the model while flow velocity amplitude is larger in the experiments. Secondly, bores form in the experiments during both flood and ebb phases. In both model and experiments the tidal flow at the sealevel boundary transitions from currents without water level fluctuations, to water level fluctuations without current fluctuations at the closed landward boundary.

## 5.8 Hypotheses for differences between measured and modelled velocities

There are minor differences between modelled and measured velocity in the tilting setup during slack, suggesting a phase difference. A possible reason is that the water depth varies with about 5 mm in the experiments over the tidal cycle, which changes inertia, whereas the modelled water depths show no significant temporal variation. In the flume there are stilling basins from which water flows in at nearly zero velocity to rapidly accelerate into the flume. In the model, on the other hand, there is no velocity gradient at the boundaries. An alternative explanation is the effect of the critical flow over the weir and the capacity of the pumps. During inflow, the water depth at and near the boundary reduces as the pump capacity is constant because less water flows out of the flume. This reduces the inflow velocity. This effect could in future be removed by increasing the pump capacity or decreasing the effective width of the channel.

During outflow, the water depth at and near the boundary increases as the broad-crested weir forces flow to be critical. This adverse behaviour could in future be removed by compensation of the weir elevation. On the other hand, inflow velocity appeared to be limited in the sand-bed experiments too, which had a considerable water volume in the 'sea' that should have

buffered inflow limitations. This suggests that the pumps are not limiting the inflow after all. We speculate that the inflow from the stilling basin (and sediment trap) over a sharp edge onto the grass-covered flume floor causes flow losses.

## 6 Results of scaling by analysis and modelling

The shallow flow equations used here are well-known to reproduce tidal dynamics in idealised tidal basins at prototype scale
(Lanzoni and Seminara, 2002; Friedrichs, 2010), and were shown above to reproduce tidal dynamics reasonably well in tidal basins at the experimental scale for both the Reynolds setup and the tilting setup. In this section we apply the model to length scales ranging from small experiments to the largest estuaries on Earth in order to explore scale effects and develop understanding how to upscale future results of live-bed experiments to natural scales. Here, scale is defined as $n_L = L_{\text{prototype}}/L_{\text{Metronome}}$. We proceed in the opposite direction of usual scaling analysis: given the Metronome we explore what systems in nature are similar to it in important properties. The consequences for scaling, expressed in characteristic dimensionless numbers for tidal systems, are discussed on the basis of modelled velocities and compared to data of real systems and some experiments (Table 2). Furthermore experiments in smaller setups than the Metronome are discussed.

### 6.1 Application of the numerical model to scale up to prototype systems

As the most critical test we assumed no distortion but simply multiplied length, width, depth, tidal amplitude and tidal period with scale factors 0.1 to 10,000, covering small 2 m long flumes to 200 km long estuaries. The results are hypothetical dimensions and dynamic properties of estuaries at the full natural scale that are geometrically the same the experiments (Table 2). The modelling is conducted to investigate whether the tidal flow is similar as well. Comparison between the hypothetical and real natural estuaries in the next section on the basis of modelled and observed velocities will show whether the assumed scaling is realistic.

The main scaling requirement for morphodynamic similarity between experiments and reality is that sediment mobility is the same regardless of the length scale. Here the grain-related Shields number is calculated from the depth-averaged flow velocity and skin-friction as $\theta = \rho u^2/[C_b^2(\rho_s - \rho)D]$. Furthermore the relative excursion length $L_e/L$ should be similar and flow should be subcritical ($Fr < 1$) which is not an arbitrary requirement in small experiments (Kleinhans et al., 2014a). The Reynolds setup is limited by the relative tidal amplitude $a/h$. The tilting setup drives tidal flow by the pressure gradient along the flume, meaning that a different scaling is required: the pressure gradient depends on water depth, which is linearly scaled with length, and gradient, which should therefore remain constant and be independent of the scale number. Tides are generated in the Reynolds setup with $a = 0.01$ m at the seaward boundary, scaled with $n_L$, and in the tilting by $a = 0.1$ m tilting amplitude that is kept constant across scales. Other model settings at the experimental scale are $W = 1.5$ m (no convergence), $h = 0.03$ m, while $C = 11 \sqrt{m}/s$ is scaled by $n_L^{1/6}$. Note, however, that in live-bed experiments these are dependent variables.

Modelled velocities and sediment mobility in the tilt2 runs remain approximately the same across all scales because the scaling is entirely linear in $n_L$ except for $C$ and for the tilting amplitude (Fig. 16). Ebb or flood dominance along the system is negligible. This means that with two open boundaries the flume simulates a reach within an estuary with a nearly static

water surface and periodic flow velocity, that is similar to the rigid lid assumption in tidal bar theories (Schramkowski et al., 2002; Leuven et al., 2016). The tilt1 runs are similar except for the surface amplitude increase and velocity amplitude decrease towards the landwards closed boundary. Further model tests (not shown) suggest that convergent planform shapes could compensate the friction loss. The Reynolds runs have a landward decaying velocity amplitude with flood dominance in flow

velocity and a longer ebb-duration landward of the mouth region. Moreover, the velocity amplitude of the Reynolds setup rapidly reduces at experimental scales while it becomes independent of scale for the largest cases. For the Reynolds setup at prototype scale sediment is in motion along nearly the entire estuary, whereas at the experimental scale the mouth region has barely mobile sediment. At the smallest scales the tidal wave fits a number of times in the basin whereas the largest scales have short basin properties. Increasing the tidal amplitude in the experiments much beyond the present $a/h = 0.33$ is not an option,

and neither is reducing the tidal period because then the tidal wavelength would become smaller than the flume length while estuaries typically have lengths less than half the tidal wavelength.

     The ratio of peak flood and peak ebb flow velocity indicates whether tidal basins are respectively importing or exporting sand. The ratio of flood and ebb duration, on the other hand, indicates the tendency for mud sedimentation at slack tide. Here, flood duration is defined as the period that flow was landwards and the ebb duration as the period that flow was seawards. The

tilting setups have nearly symmetrical tides, whereas the Reynolds setup has flood-dominant conditions except near the mouth (Fig. 16). Consequences for morphodynamic experiments and possibilities to control tidal asymmetry are discussed later.

### 6.2    Comparison of dimensionless numbers across a range of scales in idealised and real systems

The question is now how the models at different scales, and experiments, from literature and the tilting setup, compare to natural estuaries. Dimensional and dimensionless variables are given in Table 2.

Comparison between the experimental and natural estuaries shows the expected differences in flow depth, flow velocity, sediment mobility and roughness. For prototype scale, the assumed roughness in the model is similar to that in real estuaries, but the depth is considerably larger. This could suggest that the chosen depth in the model at the experimental scale is large relative to the length and width, but this remains to be investigated in live-bed experiments where the roughness is probably smaller as the sand-bed experiments had a two times larger Chézy value than the grass floor experiments. Smaller depth and the

concurrent smaller water surface amplitude would reduce sediment mobility in the Reynolds setup even further. The Froude number is below unity at the experimental scale, but is much larger than in nature. A smaller depth would increase the Froude number, but $Fr = 1$ would not be exceeded on a mobile bed (Kleinhans et al., 2014a). The width is rather arbitrarily chosen here and leads to smaller channel aspect ratios than observed in nature, which agrees with the suggestion that the modelled depth is rather large. However, in nature the channel width is a property determined by antecedent geology, by tidal prism and

by the strength of the banks. In turn, the width determines bar dimensions and bar pattern, suggesting that an interplay between self-formed bank properties and channel dimensions similar to that in rivers. Investigating this further requires morphodynamic experiments with salt marsh and riparian vegetation.

     The different wave behaviour between scales is caused by the assumed linear dependence of $L_t/L$ on scale: tidal wave celerity depends on $h^{1/2}$ while time is scaled linearly in our scenario. The tidal wave propagates in the Reynolds setup but is

closer to standing in the tilting setup. The depth in the 200 km long model scenario is 300 m, which is unrealistically deep, but again note that our chosen depth at $n_L = 1$ is not the result of morphological experiments but merely a first estimate that can be adjusted on the basis of morphodynamic experiments. The Reynolds setup is flood-dominant landward of the mouth whereas the tilting shows no significant asymmetry.

Friction dominates over inertia except in the Reynolds experiments. This means that a linear upscaling of the tilting Metronome works for estuaries of the size of the Dovey estuary and perhaps larger estuaries (Table 2). However, in smaller experiments such as those of Kleinhans et al. (2014b) (Table 2) and the 2 m scale in the model runs (Fig. 16) the friction dominates much more over inertia, while advection is also more important than in the larger flume (Table 2). Since all these numbers depend on the flow velocity, tidal period and length of the flume, they show that a flume larger than at least 10 m

length is required to obtain acceptably similar conditions to natural systems. While useful exploratory work can be done in a 2– 3 m tilting flume, similarity in tidal behaviour, flow conditions and sediment mobility is less satisfactory. All this is ultimately the consequence of having to use relatively coarse sediment to prevent cohesion and adverse hydraulically-smooth boundary effects.

The horizontal dimensions of a simulated estuary can be expressed as tidal wavelength relative to basin length and relative

tidal excursion length. Here, the tidal excursion length is the distance that a parcel of water travels during half a tidal cycle. In a natural estuary the tidal excursion length is $O(10^4)$ m, several times shorter than the estuary, while the tidal wavelength is an order of magnitude larger, and several times longer than the estuary. The same is the case for the experiments, which was an important scaling requirement. However, both the tidal excursion length and the tidal wavelength are large relative to the flume length meaning that these simulate only part of the estuary length. Reducing the tidal tilting period would reduce tidal

excursion length and tidal wavelength linearly.

## 7   Discussion: implications for morphodynamic tidal experiments

The key result of the experiments and numerical modelling is that the periodic flow velocities in a tilting flume setup are roughly uniform along the tilting flume. In contrast, periodic fluctuation of the sealevel as in Reynolds setup causes much lower velocities that decay rapidly in the landward direction and are too small to move sand. The effect on sediment transport

would be that sediment is immobile along most of the Reynolds setup and mobile in the tilting setup (Fig. 15). The ongoing morphological experiments (Fig. 2) are also conducted with this sediment. Peak values of the Shields number in the tilting experiments with grass bed are 0.2–0.3 and in the sand-bed experiments, with lower tilting slope amplitude, mobility was kept deliberately at about the threshold for motion. Much larger Shields values can be obtained by higher tilting slopes. This means that the typical mobilities of natural systems, which are $O(1)$, are within reach with the Metronome. On the other hand, to

approach such conditions in the Reynolds setup, a flume of at least 100 m length is needed with a pumping capacity of several $m^3/s$. At a length of about 200 m the flow conditions of the Reynolds setup and the tilting setup are similar (Fig. 16), while for smaller flumes only the tilting setup maintains sufficient sediment mobility.

The width and depth of self-formed estuaries depends on the strength of the banks, and in turn the bar pattern depends on the channel aspect ratio. Experimental creation of intertidal mud flats and salt marsh requires slightly-cohesive mud simulant and vegetation with flow resistance and rooting depth, appropriate at the experimental scale and in practice constrained by the typical minimum sizes of fast-sprouting vascular plants (Kleinhans et al., 2015a). This means that fine sediments, perhaps of
lower density than sand, and seeds will need to be suspended up onto the bars in tidal experiments. However, major complication in past meandering river experiments was the rapid drop of sediment mobility and of turbulence on shallower parts of the bed. Indeed, the Reynolds numbers in the smaller experiments are close to the transition from turbulent to laminar flow (Table 2) and are far below it in the smallest experiment for width-averaged flow conditions. As both flow velocity and water depth decrease onto bars, while the roughness length remains the same, the Reynolds number decreases quadratically with
depth. Therefore flow on bars is likely laminar in meter-scale experiments. This problem is considerably reduced for the 20 m flume, but not entirely removed. The 20 m flume will also be more appropriate for the use of live plant seedlings to simulate vegetation, where hydraulic resistance is Reynolds-number dependent and the rooting length relative to channel depth is more similar than in experiments in 2 m flumes.

     The Reynolds setup is slightly flood-dominated compared to the tilting setup (Fig. 16), suggesting that Reynolds experiments
would import sediment but the tilting setup would not. These results led us to hypothesise that tidal asymmetry can be imposed at any degree by asymmetric tilting by adding an overtide, which is confirmed by the model (Fig. 17). This is, for example, similar to having an M4 tidal component at the seaward boundary as occurs in nature in shallow seas such as the North Sea. To show whether the higher peak flood velocity or the longer the ebb duration dominates net transport, we calculated sediment transport from $q_s = \alpha(\theta - \theta_c)^{1.65}$ (Ribberink, 1998) and cumulated over two tidal periods. In this example, an overtide tilting
magnitude of 20% of the principal tide and a phase delay of $\pi/2$ is predicted to give strong flood-dominated transport. The transport without the imposed overtide is not exactly symmetrical because of the secondary overtide generated in the flume (Fig. 8).

     It is technically straightforward to tilt the Metronome with higher harmonics and tidal asymmetry (Fig. 17), to add a constant discharge at one closed boundary, and to impose any initial planform shape and depth along the system. This opens up possi-
bilities to drive ebb- or flood-dominant transport in the flume, which is the cause of sediment import, export and equilibrium in natural systems (Dronkers, 1986; Wang et al., 2002; Schuttelaars and de Swart, 2000), and simulate, for example, the infilling of flood-dominated estuaries. The broad similarity between conditions in the 15, 30, 40 and 60 s experiments and further model tests (not shown) indicates that a range of combinations of tidal wavelengths, tidal excursion lengths and sediment mobility can be attained in the Metronome to design preferred scales. We expect that self-formed morphology will not cause such
strong spatial variations in flow velocity as observed in the present experiments and models, because such flow divergence in a friction-dominated flow would cause spatial gradients in sediment transport that modify the morphology to reduce eventually the spatial variations in flow velocity (Savenije, 2015).

     It is clear from the present results that obtaining sufficiently mobile sediment over the length of an experimental estuary is impossible in the Reynolds setup at practical laboratory sizes. On the other hand, by tilting we have a high degree of
control over current velocities and water levels and tidal asymmetry. With this, the question whether flooding or tilting is

better suited for morphodynamic experiments of tidal systems is partially answered in that the tilting method is clearly more suited to obtain periodically reversing sediment transport at any required mobility and tidal asymmetry similar to that in natural systems. Moreover, this technology is potentially widely available because of the simplicity of periodic tilting and the smallest, meter-scale flume size at which interesting results are obtained despite serious scale problems of shallow flow (Kleinhans et al., 2014b, 2015b). The Metronome setup opens up the possibility to conduct experiments on estuary development and biogeomorphodynamics following similar principles as for rivers, including bar formation and interactions with self-forming floodplains of cohesive sediment and vegetation (Kleinhans et al., 2014a).

## 8 Conclusions

The prime requirement for scale experiments of tidal systems is to obtain reversing currents that cause high sediment mobility along the entire system, while tidal wave behaviour is of secondary importance. However, it is impossible to scale the sediment size by the same factor as system length and width. Here we show that the degree of similarity of tidal flows and sediment mobility in experiments and in nature depends strongly on the size of the flume and the method of generating tidal flows.

The method of Reynolds with periodically fluctuating sealevel cannot lead to sufficient bed shear stress for bidirectional sediment transport except with low-density sediments and in impractically setups larger than hundreds of meters. The reason is that the tidal wave rapidly dampens out in landward direction due to friction, which is higher in experiments than in nature and much higher in flumes of a few meters long. This cannot be compensated by higher tidal amplitude because that should not exceed half the water depth. A practical problem is that these experiments require considerable pumping capacity even to reach the limited mobility in the inlet.

A periodically tilting flume of 20 m length causes reversing flows with sufficient strength in both flood and ebb direction to transport sand. A sinusoidal tilting pattern with two open boundaries causes an approximately sinusoidal flow velocity pattern along the entire flume with uniform width and depth whilst water level hardly fluctuates. This means that the rigid lid condition is approximated with two open boundaries. When one boundary is closed, reflection of the tidal wave causes large depth fluctuations and enhanced ebb currents near the closed boundary, whilst the flow velocity along most of the flume is almost the same as in the experiments with two open boundaries. The flow remains subcritical but the Froude number is much larger than in natural estuaries. In nature this would affect tidal wave propagation and resulting flow velocity, but in the tilting flume the tidal wave is independently imposed by the tilting. The flow is turbulent in the present experimental conditions, but the Reynolds number would drop rapidly to laminar conditions for shallower flows above bars, meaning that larger flumes are better for suspended sediment transport onto bars and mud flats and for interactions with live seedling vegetation.

Numerical modelling for a range of scales shows that similar sediment mobility, relative tidal excursion length and relative tidal wavelength can be attained in tilting flumes of tens of meters long. The tidal flow is friction-dominated, as in natural systems that are three orders of magnitude larger, while advection is of minor importance. However, for the smallest possible experimental estuaries of a few meters length the friction is much higher than in nature while the flow becomes laminar above bars, although the required sediment mobility may be attained which is useful for exploratory experimentation.

The tilting flume setup allows independent control over tidal period and tidal asymmetry. This, in turn, allows experimental control over the simulated length of the tidal basin, tidal excursion length and tendency to import or export sediment without compromising the sediment mobility. The implication is that the Metronome tidal facility opens up new possibilities for tidal morphodynamics research that are complementary to numerical modeling and field observations.

5 *Author contributions.* The authors contributed as follows: idea of flume tilting and principal investigator: MK, data collection: MK, JL and LB, data analyses: MK and JL, numerical model development: MV, numerical modelling and scaling: MK and MV, manuscript preparation: MK with contributions from JL, LB and MV. MvM and MK led the design and building project, HM and AS did the technical design of the Metronome, CR, HM and AE designed and built the pumping system, PV led laboratory infrastructure design and construction with MvM.

*Acknowledgements.* Comments by two anonymous reviewers, steer by Associate Editor Daniel Parsons and discussions with Job Dronkers
10 and Huub Savenije are gratefully acknowledged. The research by MK, JL and LB and the construction of the Metronome were funded as a Vici grant to MK by the Dutch Technology Foundation (STW/TTW) that is part of the Netherlands Organisation for Scientific Research (NWO, grant 016.140.316). We acknowledge collaboration with the basin design and building team at Consmema, Variodrive, who designed and programmed the actuators, Bart Boshuizen (TU-Delft) who programmed the motion, HiH engineering, who modelled the forces on the steel construction, the actuators and the floor, and Stemmer Imaging, who designed and programmed the imaging system.
15 Data of the Metronome and building plans of a 3 m long steel mini-Metronome, original images, Matlab image processing scripts and the Matlab flow model are available upon request from MK. Building plans for the Metronome are in the online supplements.

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

**Table 1.** Boundary conditions applied in all experiments: auxiliary fixed-slope experiments to determine the roughness of the artificial grass bed, periodic tilting experiments with one or two open boundaries, and periodic sealevel variations. Experiments with sand bed were conducted with a shallow sea of 2 m length to reduce boundary effects and were closed on the upstream boundary.

| bed | period $\mathrm{s}$ | tilt slope amplitude $\times 10^{-3}\mathrm{m/m}$ | sealevel amplitude $\times 10^{-3}\mathrm{m}$ | boundaries | rationale |
|---|---|---|---|---|---|
| grass |  | 0.9 | 0 | both open | steady flow: control |
| grass |  | 2.3 | 0 | both open | same, faster flow |
| grass | 60 | 0 | 10 | x=0 m closed | same, longer wave |
| grass | 30 | 0 | 20 | x=0 m closed | Reynolds method |
| grass | 30 | 9.1 | 0 | both open | reach within estuary |
| grass | 15 | 9.1 | 0 | both open | same, shorter tidal excursion length |
| grass | 30 | 4.5 | 0 | both open | reduced tidal energy (not shown) |
| grass | 30 | 9.1 | 0 | x=0 m closed | basin with reflective landward boundary |
| grass | 15 | 9.1 | 0 | x=0 m closed | same, short tidal excursion length |
| sand | 40 | 3.6 | 0 | x=0 m closed | tilting, natural roughness |
| sand | 40 | 0 | 3.5 | x=0 m closed | Reynolds method, natural roughness |

**Table 2.** Properties of Reynolds and tilting estuaries across a range of scales from experiments to natural estuaries. Linear friction is defined as $r = 8gu/(3\pi hC^2)$ with the given velocity that is usually assumed. The ratio between friction and inertia is calculated as $rT/2\pi$. Velocities for the model scenarios were taken from the model runs at relative length 0.8. Experiments with the Reynolds setup were taken from Reynolds (1889) and Tambroni et al. (2005) and with the tilting setup from Kleinhans et al. (2014b). Data of the Dovey (UK) are from Brown and Davies (2010), Thames from Friedrichs (2010) and Westerschelde from Wang et al. (2002).

| case | Metronome | Kleinhans | Fig. 2 | Metronome | Tambroni 1 | Reynolds tank A | prototype | Dovey | Thames | Westerschelde |
|---|---|---|---|---|---|---|---|---|---|---|
| | model | experiment | experiment | model | experiment | experiment | model | nature | nature | nature |
| configuration | tilt | tilt2 | tilt1 | Reynolds | Reynolds | Reynolds | Reynolds | Reynolds | Reynolds | Reynolds |
| length $L$ (m) | 20 | 3.5 | 20 | 20 | 24.14 | 3.62 | 20000 | 20000 | 95000 | 200000 |
| width $W$ (m) | 1.5 | 1.3 | 1.5 | 1.5 | 0.3 | 1.18 | 1500 | 800 | 4300 | 6000 |
| depth $h$ (m) | 0.03 | 0.004 | 0.025 | 0.03 | 0.082 | 0.05 | 30 | 5 | 8.5 | 15 |
| amplitude $a$ (m) | 0.1 | 0.018 | 0.05 | 0.01 | 0.05 | 0.05 | 10 | 2 | 2 | 1.75 |
| period $T$ (s) | 40 | 72 | 30 | 40 | 180 | 53 | 40000 | 44712 | 44712 | 44712 |
| period $T$ (hr) | | | | | | | 11 | 12.42 | 12.42 | 12.42 |
| Chézy (m$^{0.5}$/s) | 11 | 11 | 11 | 11 | 11 | 11 | 35 | 35 | 50 | 55 |
| $u$ (m/s) | 0.2 | 0.1 | 0.25 | 0.07 | 0.24 | | 1 | 1.2 | 1 | 1.5 |
| $\theta$ (-) | 1 | 0.25 | 0.5 | 0.11 | 0.33 | | 10 | 1.99 | 1.26 | 2.56 |
| $Fr$ (-) | 0.37 | 0.5 | 0.5 | 0.13 | 0.27 | | 0.06 | 0.17 | 0.11 | 0.12 |
| $Re$ (-) | 6000 | 400 | 6300 | 2100 | 19700 | | 30000000 | 6000000 | 8500000 | 22500000 |
| excursion $L_e$ (m) | 4 | 3.6 | 3.75 | 1.4 | 21.6 | | 20000 | 26827 | 22356 | 33534 |
| wavelength $L_t$ (m) | 22 | 14 | 15 | 22 | 161 | 37 | 686000 | 313000 | 408000 | 542000 |
| friction $r$ (-) | 0.46 | 1.72 | 0.69 | 0.16 | 0.2 | | 0.0002 | 0.0016 | 0.00039 | 0.00028 |
| aspect $W/h$ | 50 | 325 | 60 | 50 | 4 | 24 | 50 | 160 | 506 | 400 |
| amplitude $a/h$ | | | | 0.33 | 0.61 | 1 | 0.33 | 0.4 | 0.24 | 0.12 |
| excursion $L_e/L$ | 0.2 | 1.03 | 0.19 | 0.07 | 0.89 | 1 | 1 | 1.34 | 0.24 | 0.17 |
| wavelength $L_t/L$ | 1.1 | 4 | 0.8 | 1.1 | 6.7 | 10.2 | 34.3 | 15.7 | 4.3 | 2.7 |
| $L_t/L_e$ | 5.5 | 3.9 | 4 | 15.7 | 7.5 | | 34 | 12 | 18 | 16 |
| friction / inertia | 2.9 | 19.7 | 3.3 | 1 | 5.7 | | 1.3 | 11.4 | 2.8 | 2 |
| advection / inertia | 0.2 | 1 | 0.2 | 0.1 | 0.9 | | 1 | 1.3 | 0.2 | 0.2 |

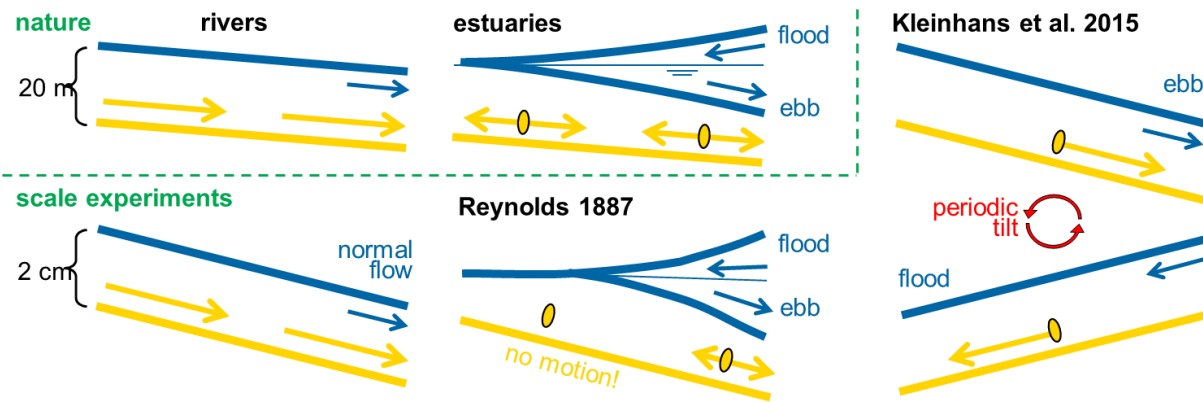

**Figure 1.** Driving flow in nature and in experiments with the requirement of sediment mobility similarity. Given the same sediments in experiments as in natural systems, the shear stress in experiments must be the same as in nature. With much smaller water depths this requires much larger gradients, which is straightforward for river experiments. In tidal experiments these gradients are impossible to obtain in flumes by sealevel fluctuation, but quite feasible to obtain by tilting the flume periodically.

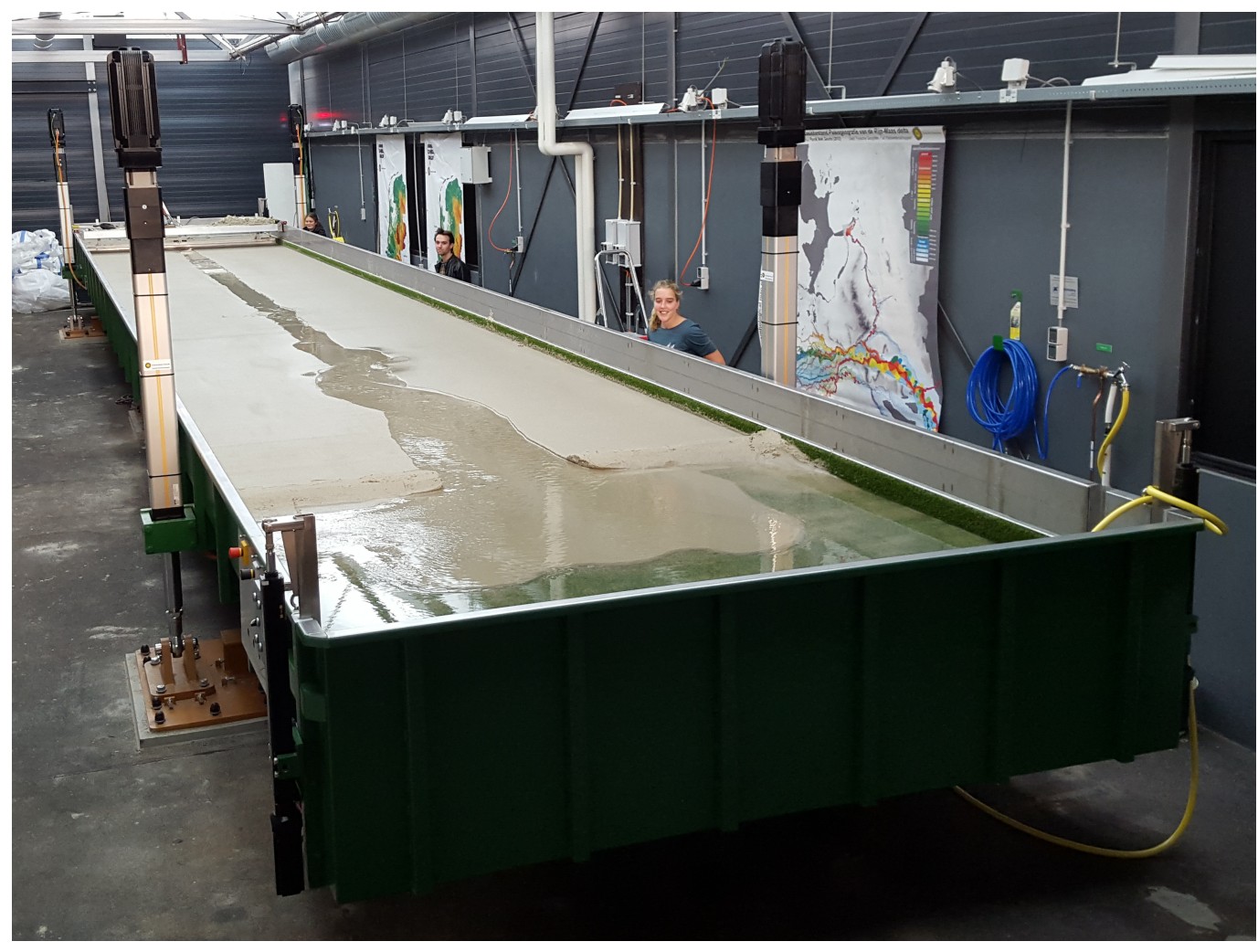

**Figure 2.** The Metronome tidal facility. Note PhD candidates for scale. The flume floor below the sand is covered in artificial grass (see text). Note the vertically mounted actuators that drive the flow. This pilot experiment started as a 0.2 m by 0.03 m straight initial channel and ran for about 12 hours with a slope amplitude of 0.005 m/m and a period of 30 s and 100 L/h river inflow.

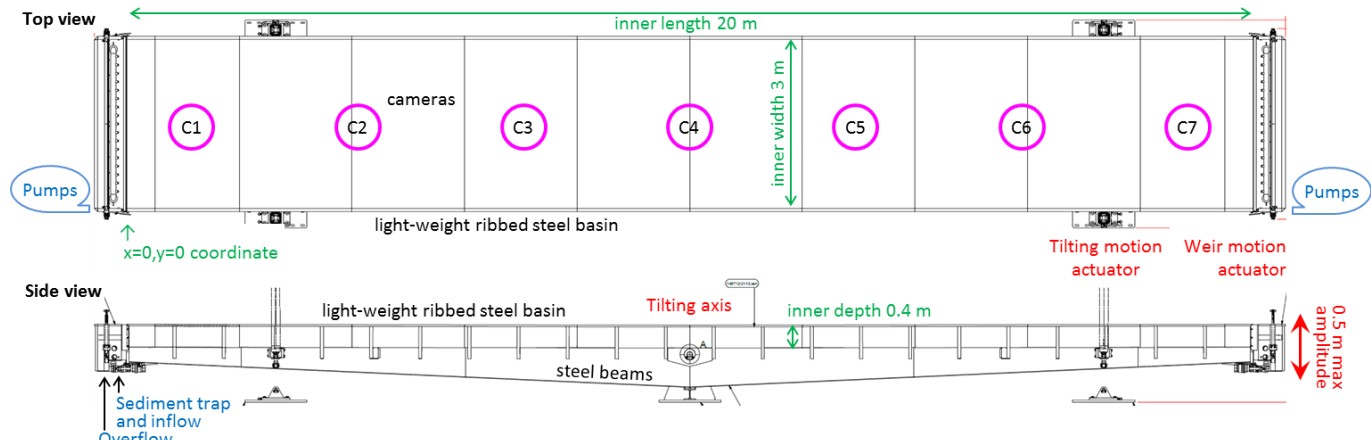

**Figure 3.** Geometry of the Metronome facility. The inner basin measures 20.00 m in length 3.00 m in width and 0.40 m in depth and the maximum tilting amplitude is 0.5 m at the end tank resulting in a tilting slope amplitude of 0.05 m/m. Both flume ends have end tanks with a 0.3 m long stilling basin functioning as sediment trap and pumped water inflow, separated by an automated weir from the outside 0.2 m long overflow basin with a 2 mm mesh to capture PIV particles. Motion is controlled by four 20 kN actuators for tilting and two small actuators for each end tank weir. Cameras C1–7 are mounted 3.7 m above the flume floor.

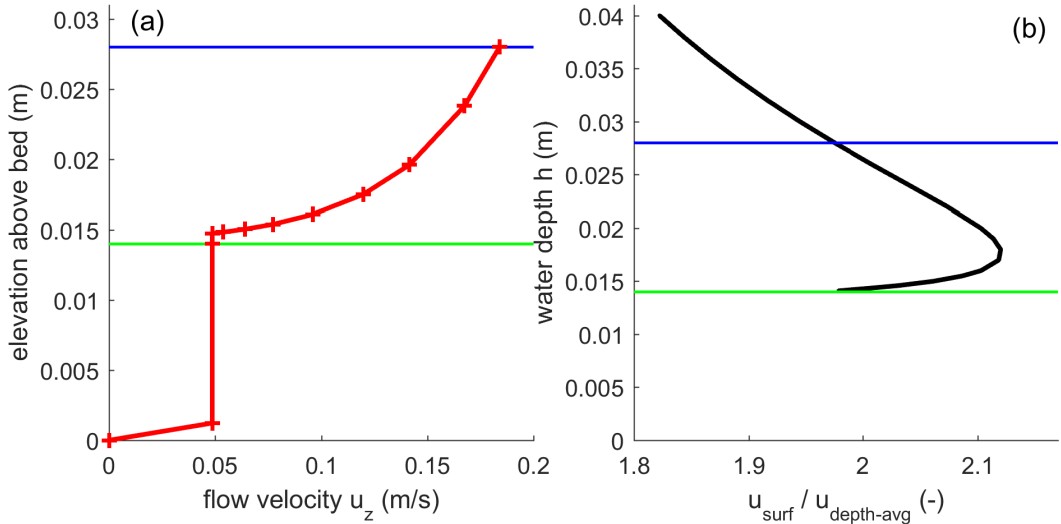

**Figure 4.** Application of the Baptist et al. (2006) vegetation friction relation to the artificial vegetation in the Metronome. (a) Flow velocity profile (red) between the bed and the water surface (blue). Height of vegetation indicated in green. (b) Ratio between water surface velocity $u_{surf}$ and depth-averaged velocity $u_{depth-avg}$ as a function of total flow depth. This is independent of slope. Given the insensitivity to water depth variations, a constant value of 1.95 is assumed for comparison of measured and modelled flow velocities.

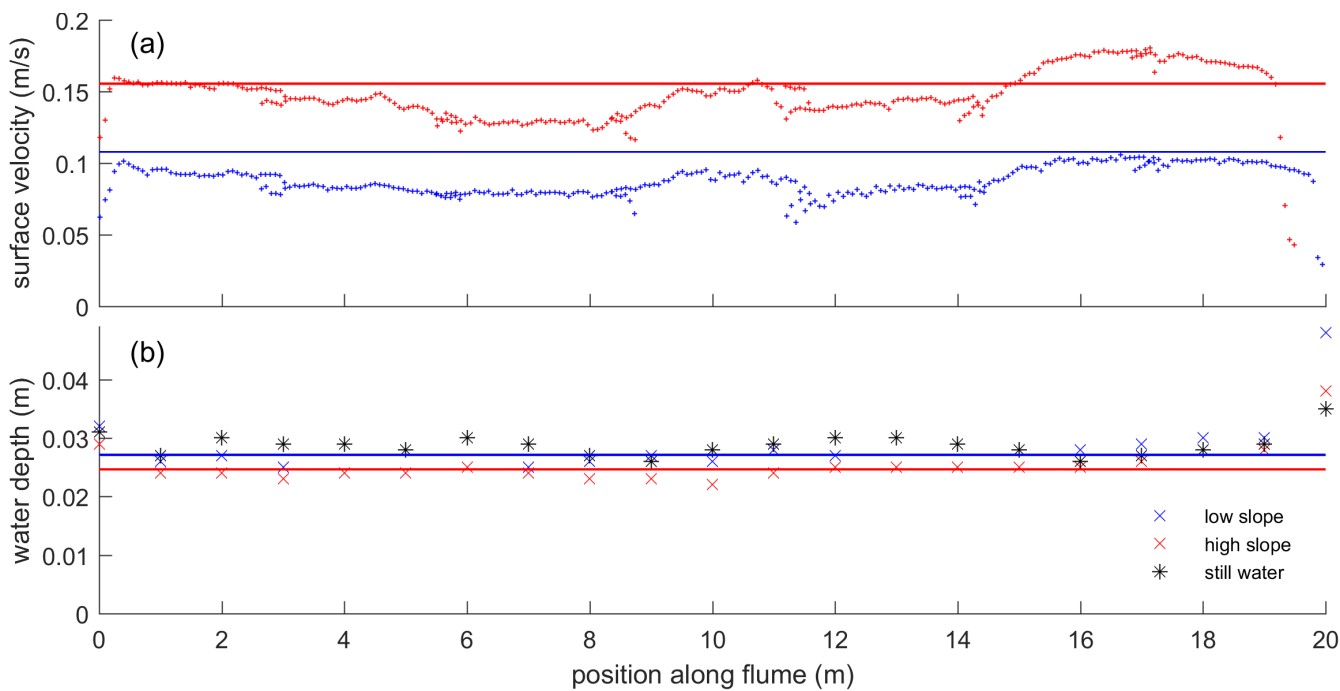

**Figure 5.** Unidirectional flow data from PIV for the constant low slope and high slope experiments. (a) Flow velocity at the water surface along the flume. Drawn lines are analytically calculated surface flow velocities. (b) Approximate water depths measured for both experiments. Drawn lines are average values used in flow calculation. Still water depth measurements show variation of bed level due to irregularities in artificial grass height and sand layer thickness at the bottom of the grass.

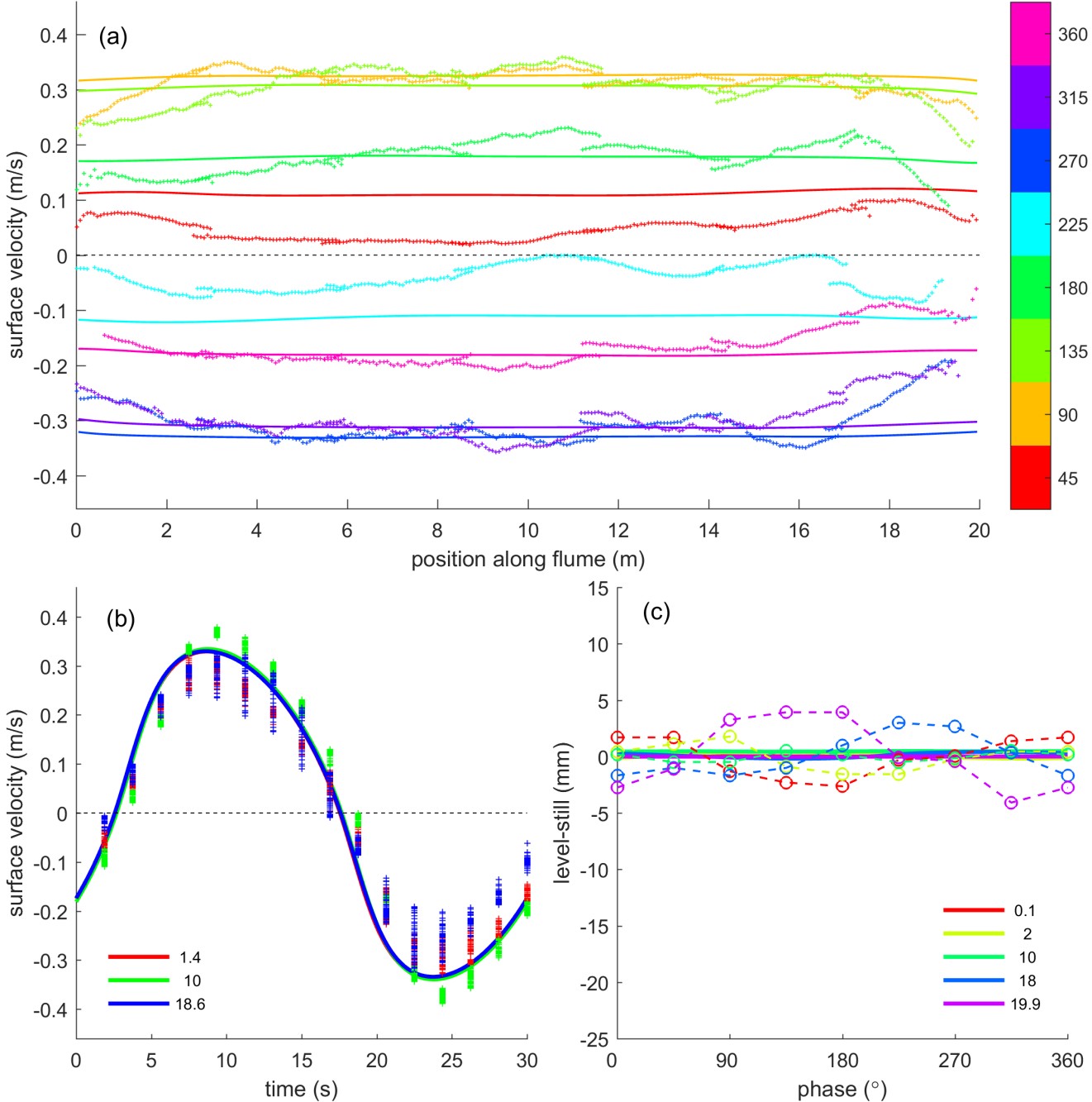

**Figure 6.** Flow data from PIV and modelled flow with 30 s period tilting at 0.009 m/m slope amplitude with both boundaries open. Flow velocity is defined as positive in the ebb-direction with $x = 0$ m being the upstream boundary. (a) Flow velocity at the water surface along the flume for selected phases of the tidal cycle. (b) Flow velocity at the water surface in one tidal cycle for selected positions along the flume measured from $x = 0$ m, indicated in legend. (c) Water level as a function of phase in the tidal cycle for selected positions along the flume. Measured water levels have correct amplitude and phase but possibly erroneous vertical offsets. Data are plotted as symbols and model results are plotted as drawn lines.

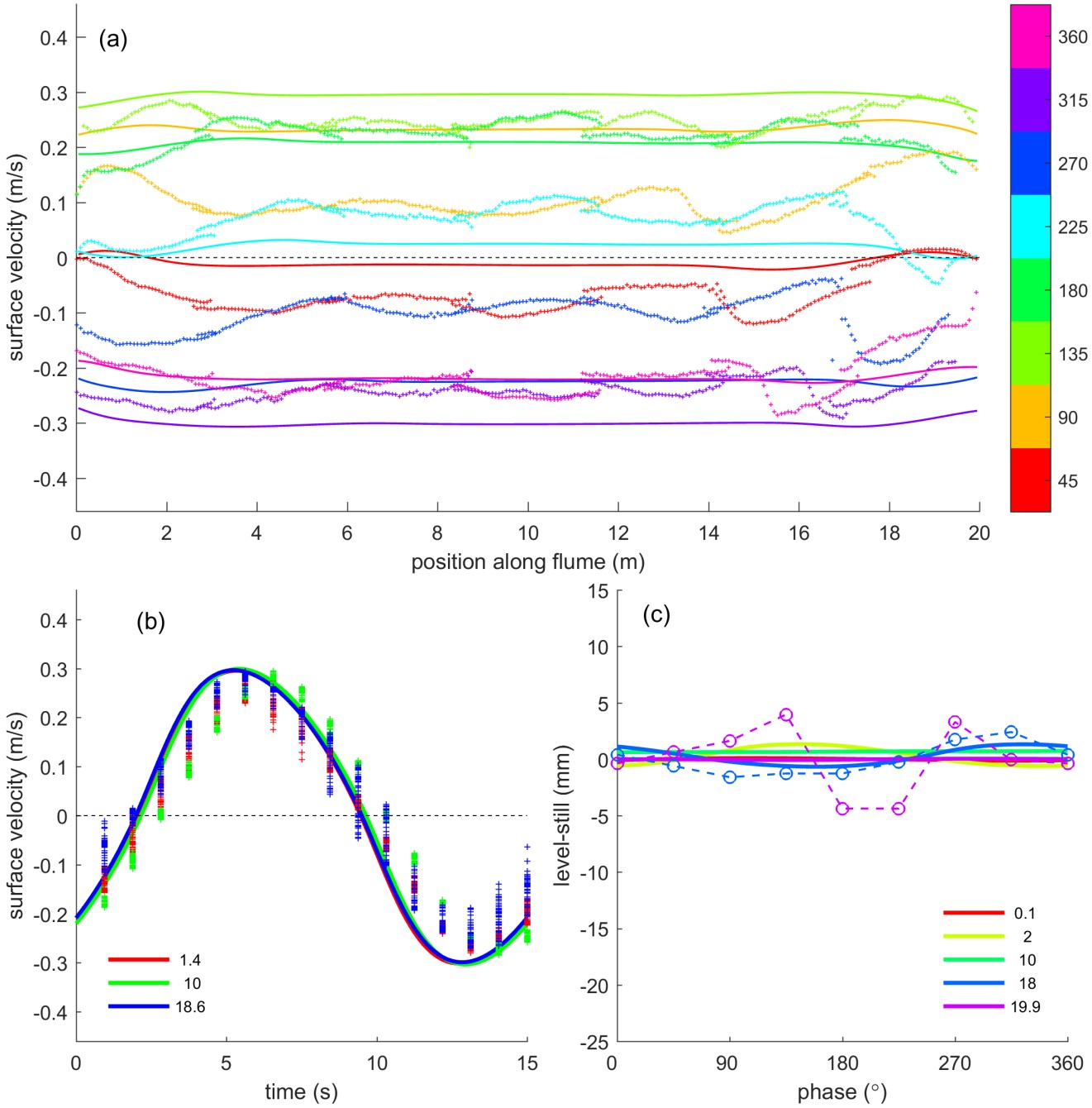

**Figure 7.** Flow data from PIV and modelled flow with 15 s period tilting at 0.009 m/m slope amplitude with both boundaries open. (a) Flow velocity at the water surface along the flume for selected phases of the tidal cycle. (b) Flow velocity at the water surface in one tidal cycle for selected positions along the flume measured from $x = 0$ m, indicated in legend. (c) Water level as a function of phase in the tidal cycle for selected positions along the flume. Measured water levels have correct amplitude and phase but possibly erroneous vertical offsets. Data are plotted as symbols and model results are plotted as drawn lines.

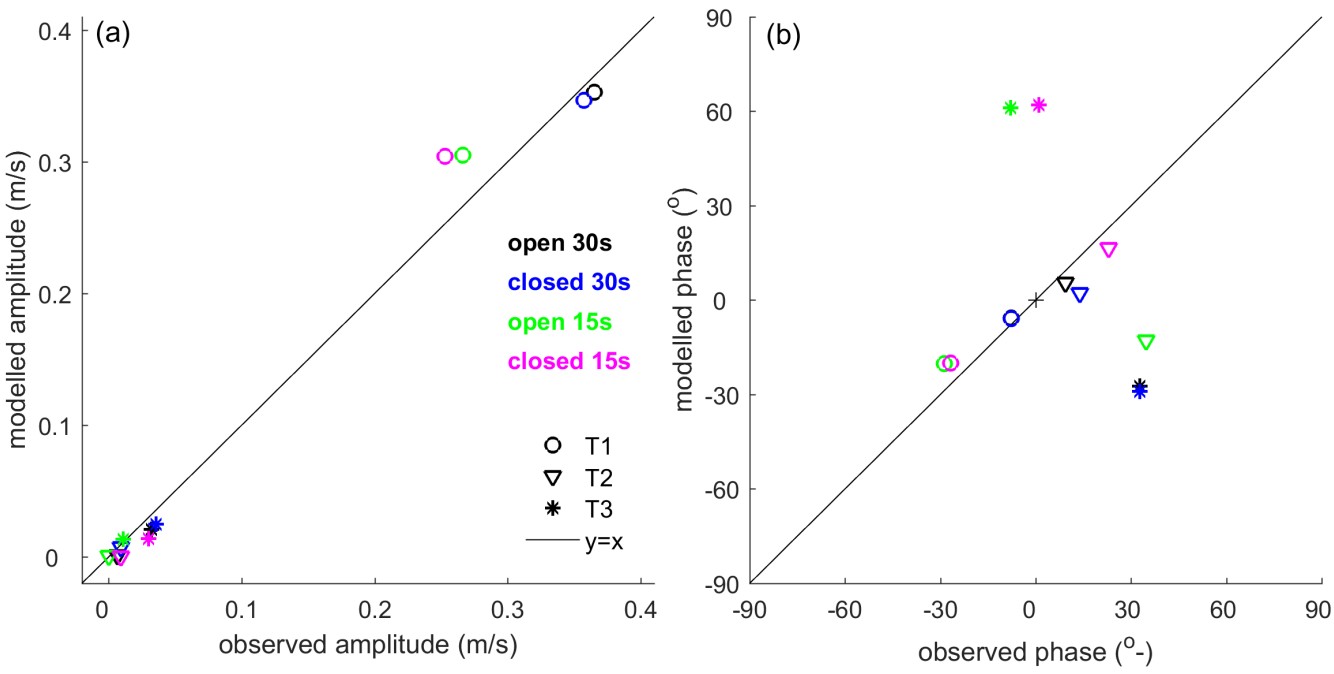

**Figure 8.** Tidal amplitude (a) and phase (b) in the velocity signal of the model runs compared to the experiments for the tilting flume. The principal tide 'T1' is the full tidal period of 30 s or 15 s and the first overtide 'T2' and second overtide 'T3' the higher harmonics. Dashed lines indicate the perfect fit plus or minus an error of 0.01 m/s to indicate the uncertainty range of the data.

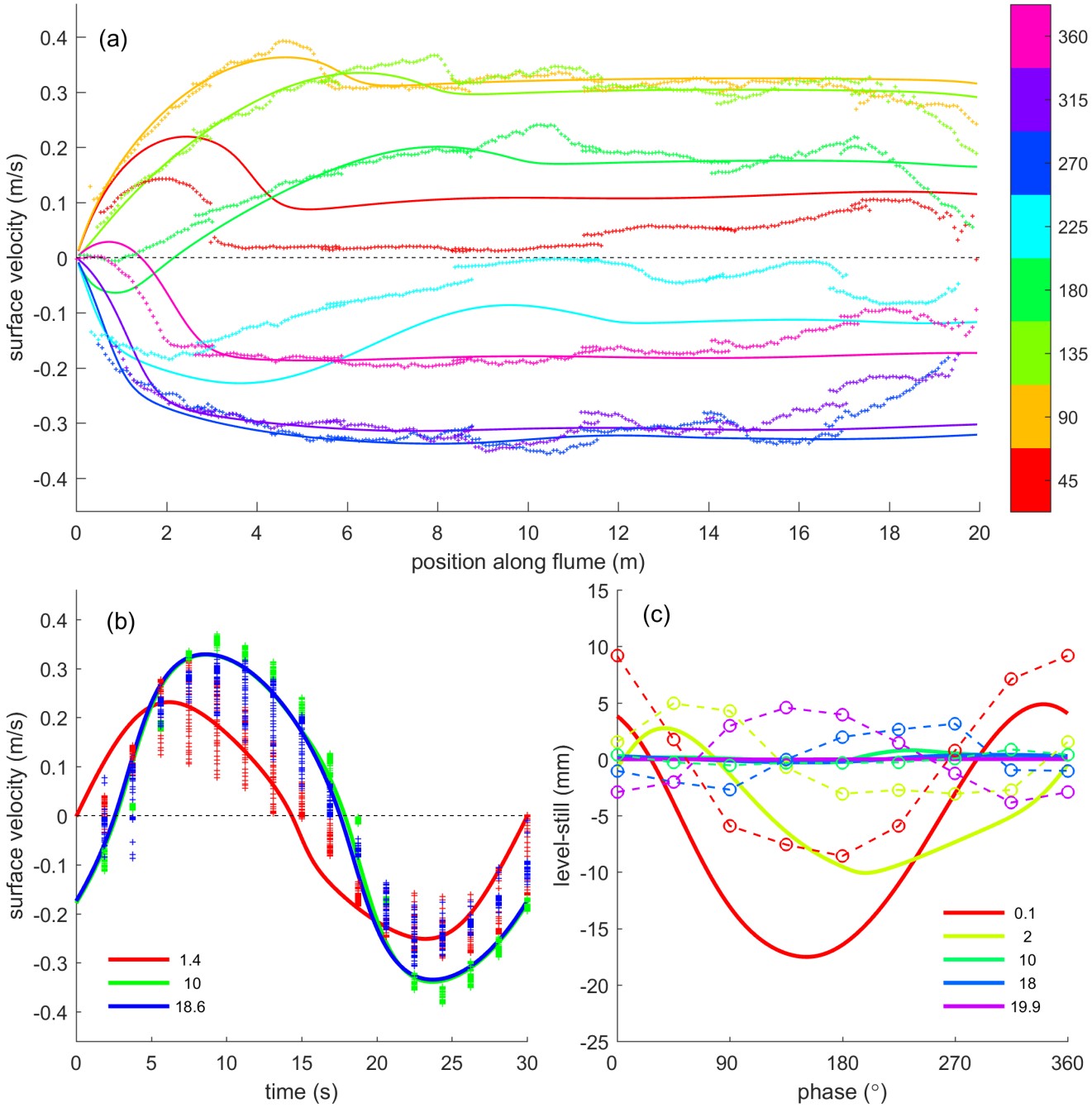

**Figure 9.** Flow data from PIV and modelled flow with 30 s period tilting at 0.009 m/m slope amplitude with upstream boundary closed. (a) Flow velocity at the water surface along the flume for selected phases of the tidal cycle. (b) Flow velocity at the water surface in one tidal cycle for selected positions along the flume measured from $x = 0$ m, indicated in legend. (c) Water level as a function of phase in the tidal cycle for selected positions along the flume. Measured water levels have correct amplitude and phase but possibly erroneous vertical offsets. Data are plotted as symbols and model results are plotted as drawn lines.

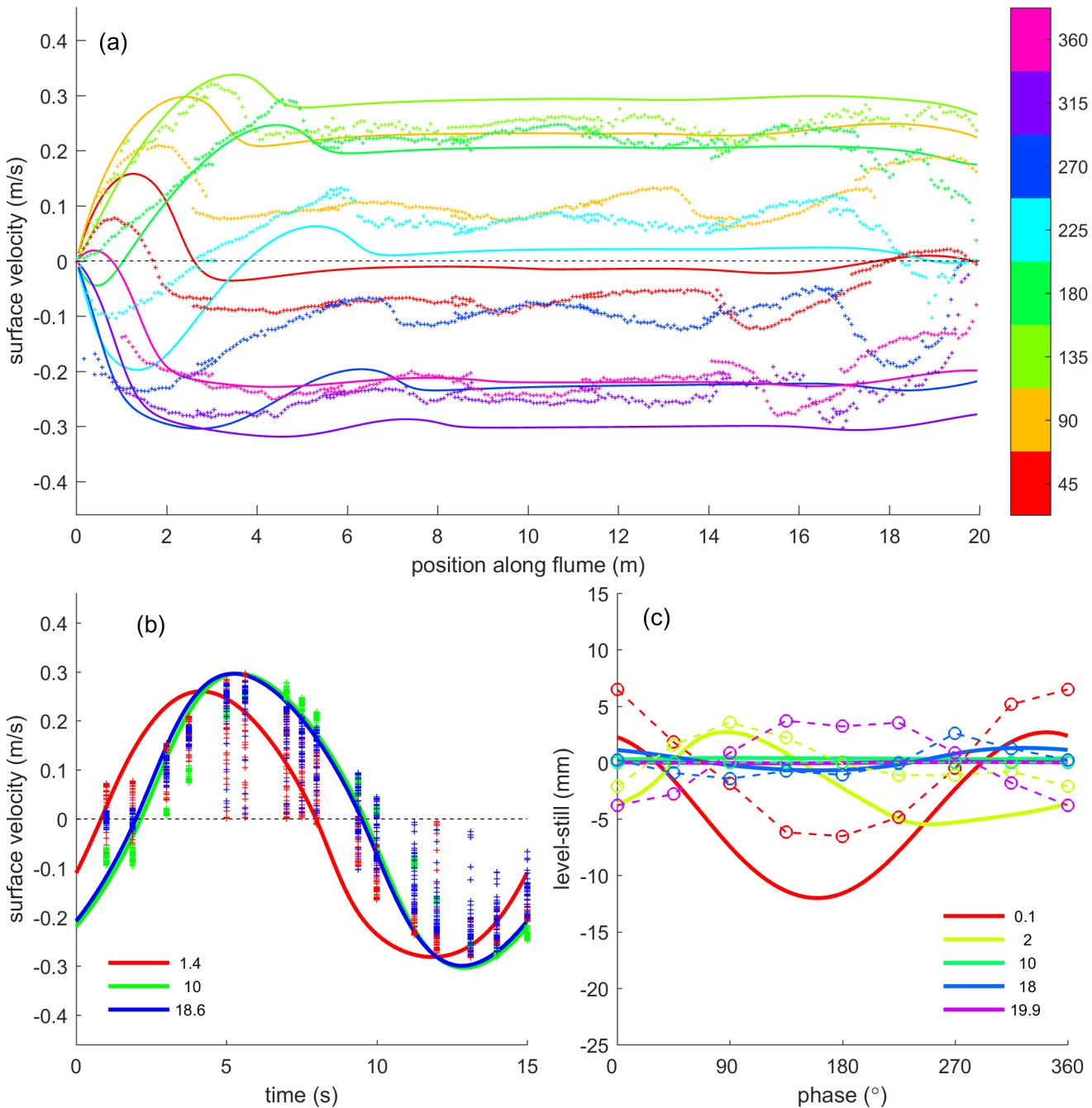

**Figure 10.** Flow data from PIV and modelled flow with 15 s period tilting at 0.009 m/m slope amplitude with upstream boundary closed. (a) Flow velocity at the water surface along the flume for selected phases of the tidal cycle. (b) Flow velocity at the water surface in one tidal cycle for selected positions along the flume measured from $x = 0$ m, indicated in legend. (c) Water level as a function of phase in the tidal cycle for selected positions along the flume. Measured water levels have correct amplitude and phase but possibly erroneous vertical offsets. Data are plotted as symbols and model results are plotted as drawn lines.

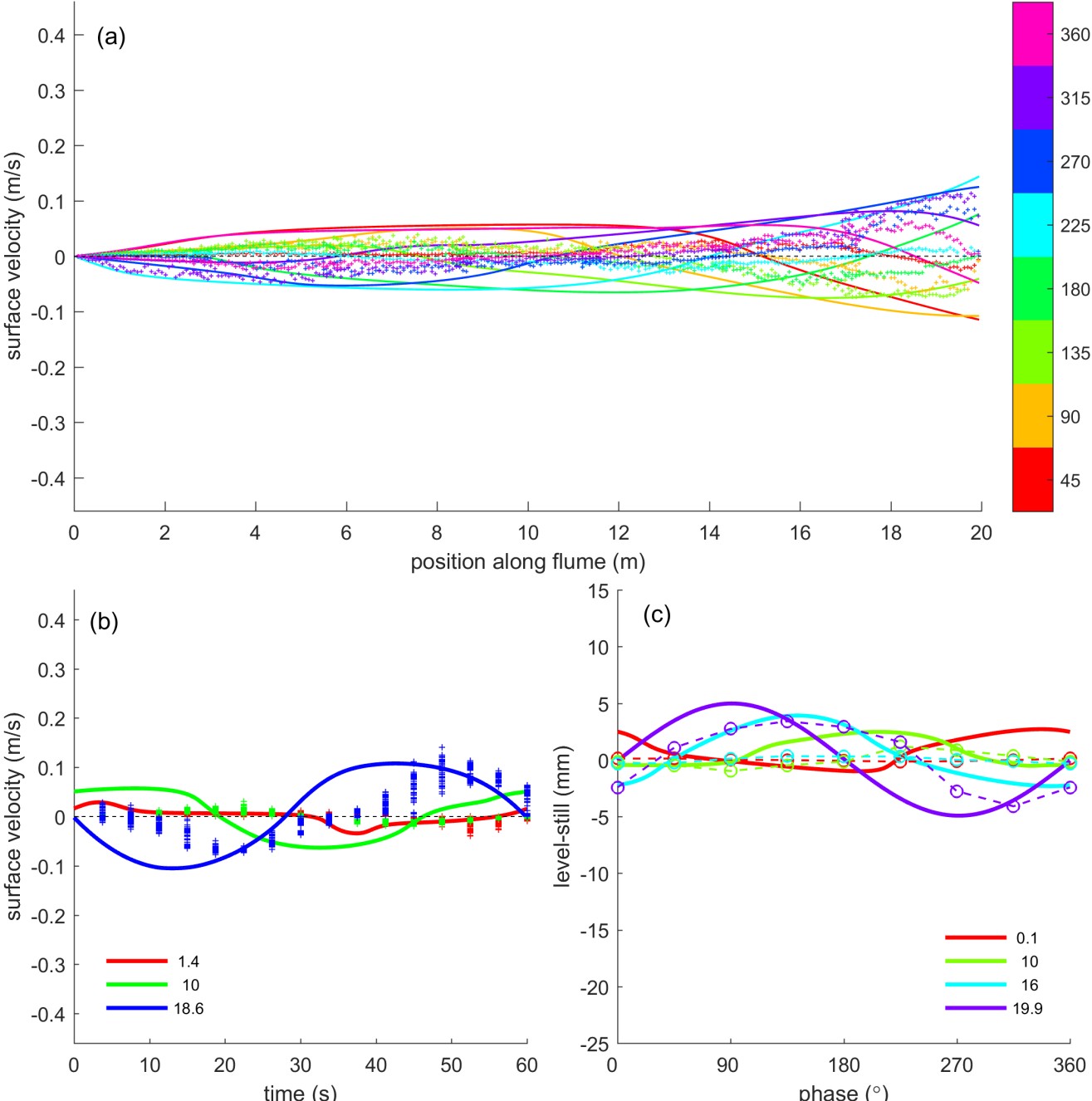

**Figure 11.** Flow data from PIV and modelled flow with 60 s period sealevel fluctuation at 0.01 m amplitude with the landward boundary closed. Positive flow velocity is in the ebb direction. (a) Flow velocity at the water surface along the flume for selected phases of the tidal cycle. (b) Flow velocity at the water surface in one tidal cycle for selected positions along the flume measured from $x = 0$ m, indicated in legend. (c) Water level as a function of phase in the tidal cycle for selected positions along the flume, indicated in the legend. Measured water levels have correct amplitude and phase but possibly erroneous vertical offsets. Data are plotted as symbols and model results are plotted as drawn lines.

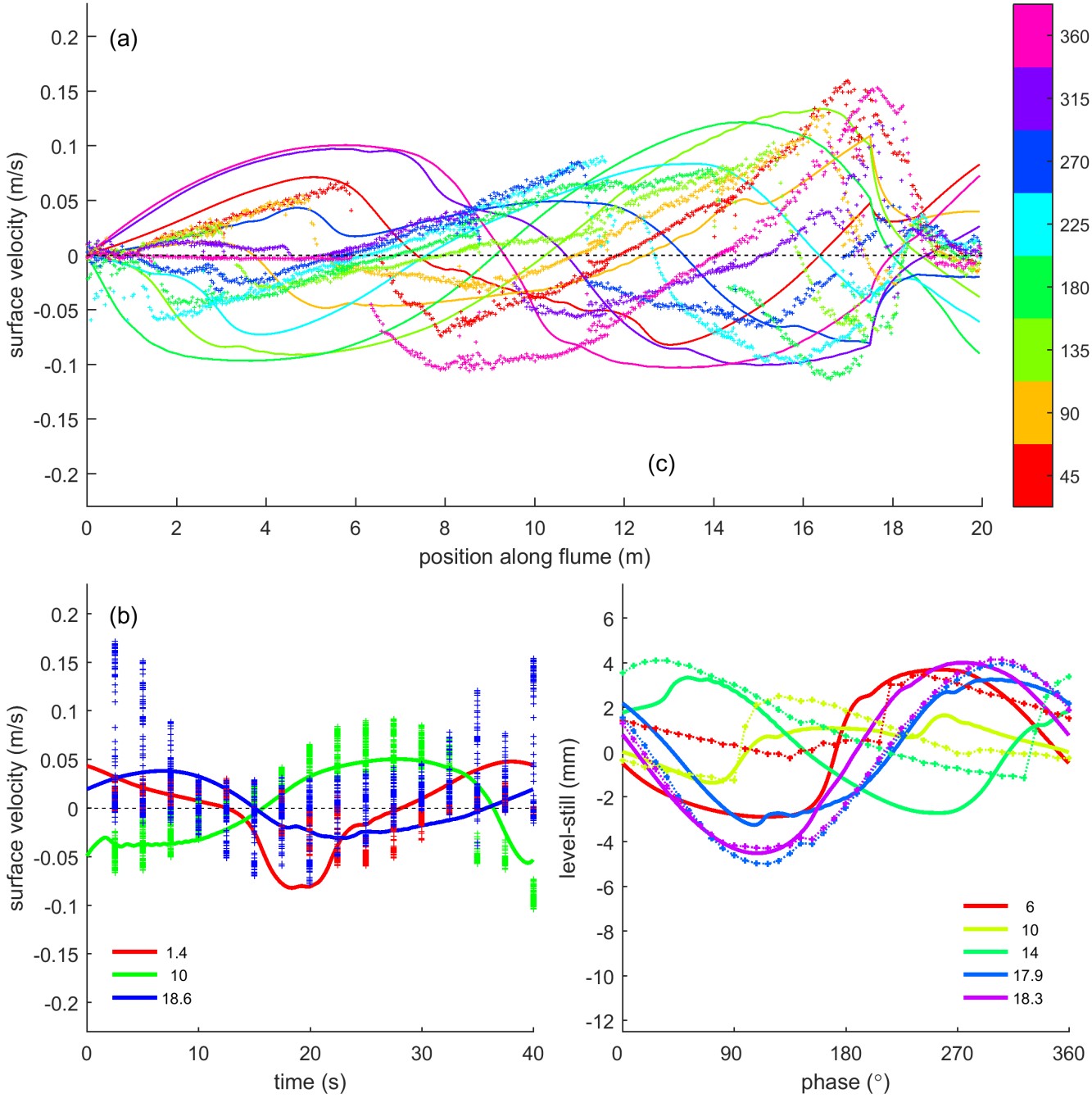

**Figure 12.** Flow data in an 18 m sand-bed channel with a 2 m long by 3 m wide sea from PIV and modelled flow with 40 s period sealevel fluctuation of 0.007 m amplitude with the landward boundary closed. Note that the vertical axis range is half that of the tilting experiment. (a) Flow velocity at the water surface along the flume for selected phases of the tidal cycle. (b) Flow velocity at the water surface in one tidal cycle for selected positions along the flume measured from $x = 0$ m, indicated in legend. (c) Water level measured by acoustics as a function of phase in the tidal cycle for selected positions along the flume. Data are plotted as symbols and model results are plotted as drawn lines.

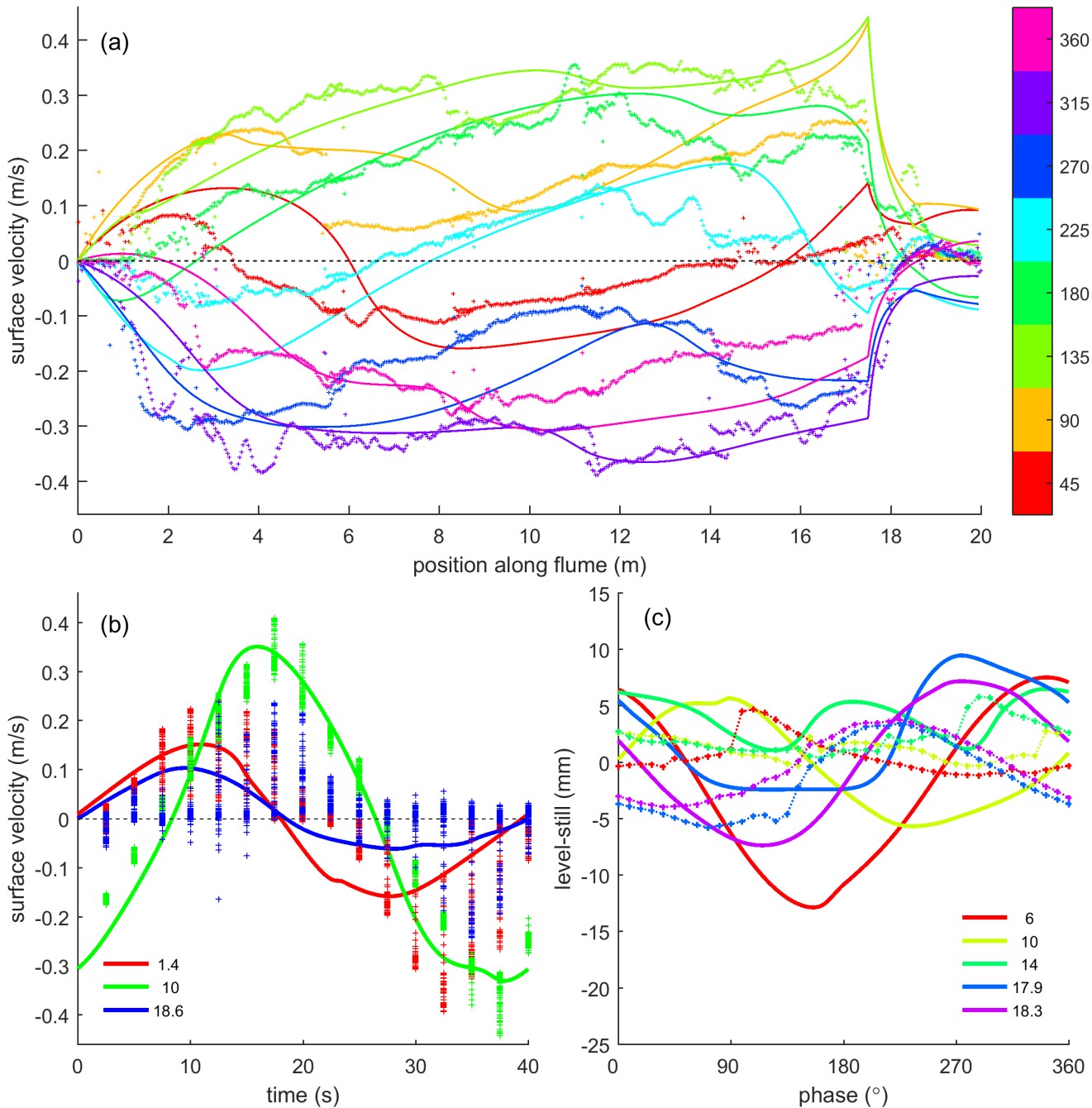

**Figure 13.** Flow data in an 18 m sand-bed channel with a 2 m long by 3 m wide sea from PIV and modelled flow with 40 s period tilting at 0.004 m/m slope amplitude with the landward boundary closed. (a) Flow velocity at the water surface along the flume for selected phases of the tidal cycle. (b) Flow velocity at the water surface in one tidal cycle for selected positions along the flume measured from $x = 0$ m, indicated in legend. (c) Water level measured by acoustics as a function of phase in the tidal cycle for selected positions along the flume. Data are plotted as symbols and model results are plotted as drawn lines.

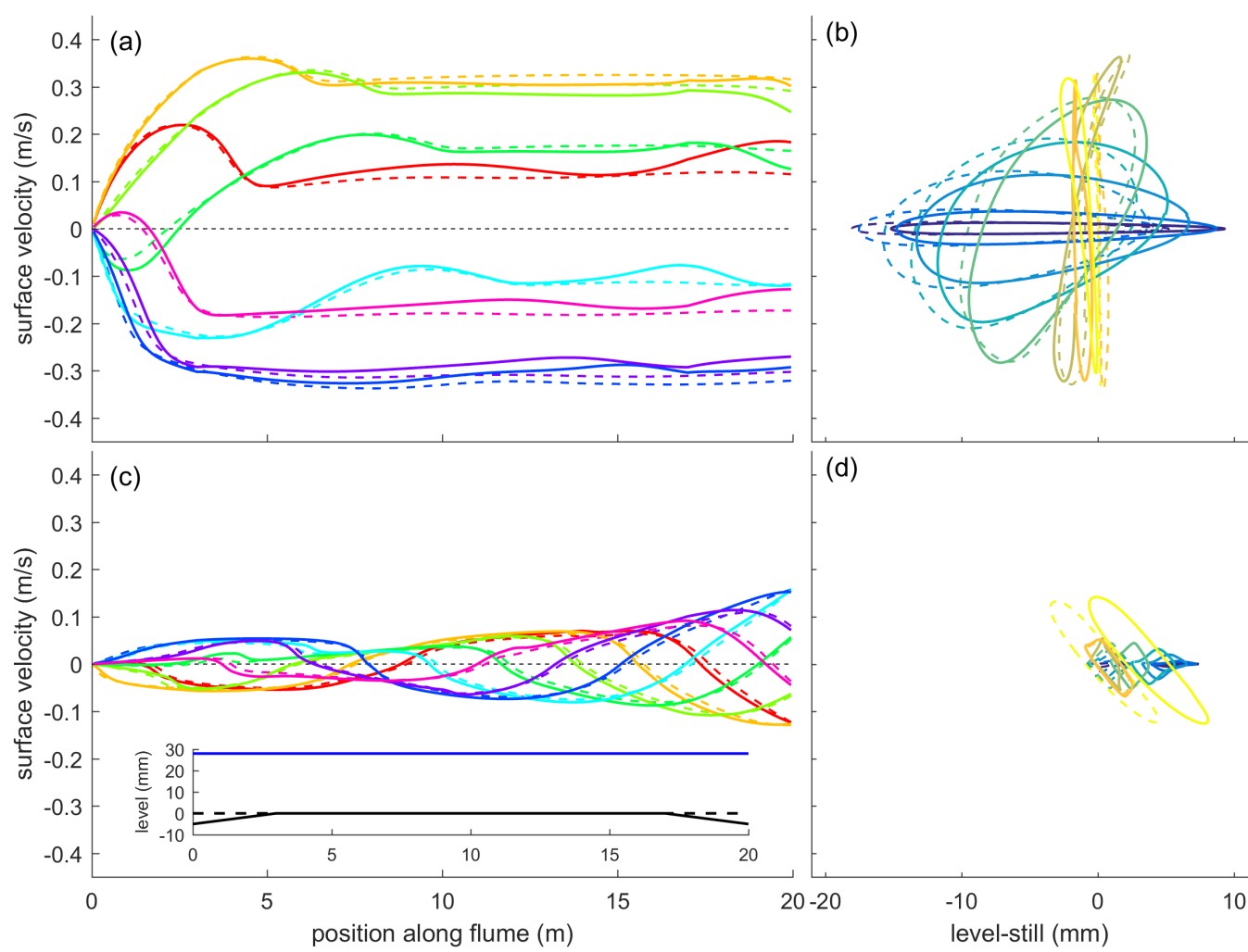

**Figure 14.** Sensitivity of flow to depth variation in the basin (drawn lines) compared with the ideal depth (dashed lines) for 30 s period model runs. Inset in (c) shows modified bed elevation. Runs with original depth are same as in Fig. 9. (a,c) Flow velocity at the water surface along the flume for selected phases of the tidal cycle. Legend as in Fig. 9. (b,d) Water level and flow velocity in one tidal cycle showing hysteresis. (a,b) Tilting at 0.09 m amplitude with upstream boundary closed. (c,d) Reynolds method with sealevel fluctuation at 0.01 m amplitude with the landward boundary closed.

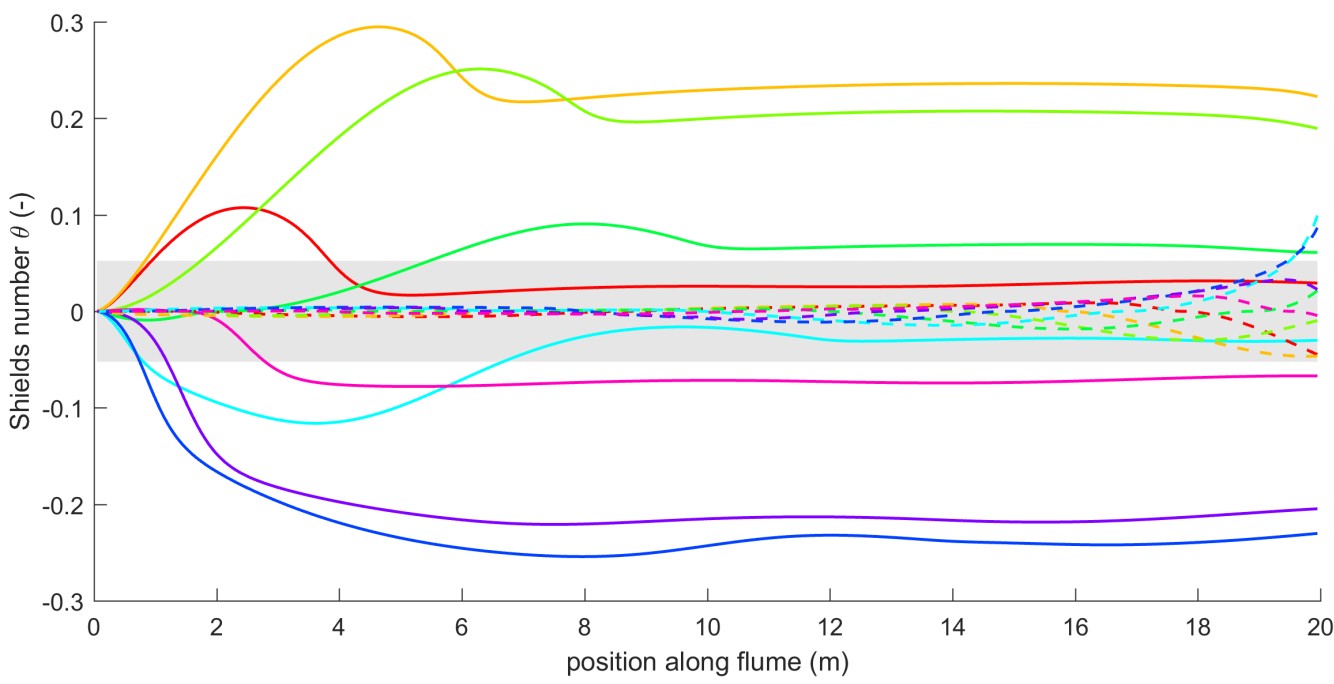

**Figure 15.** Sediment mobility calculated for tidal flows driven by periodic tilting (drawn lines) and by the Reynolds method with periodic sea surface fluctuation (dashed lines). Legend as in Fig. 9. Gray area indicates immobile sediment.

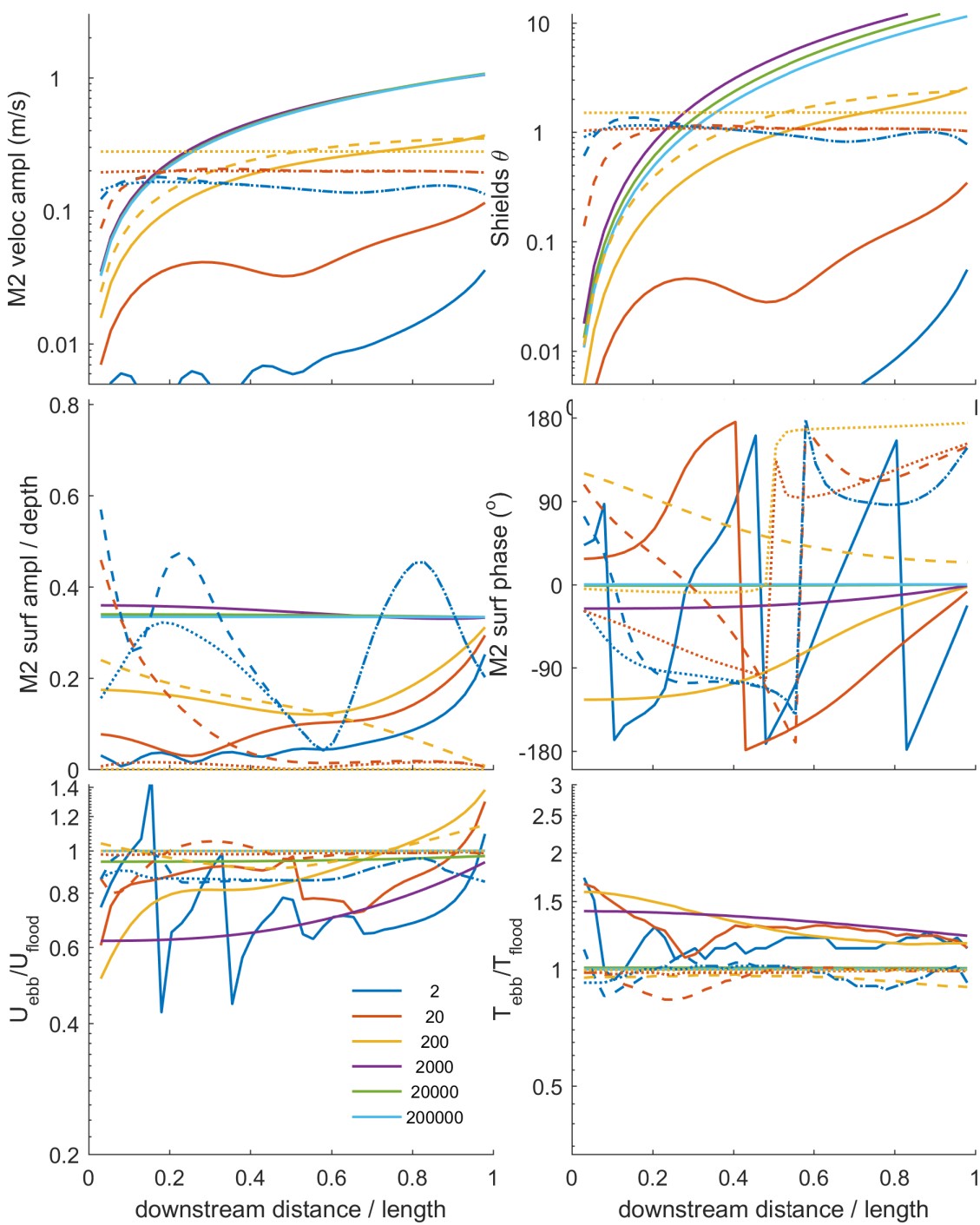

**Figure 16.** Scaling from Metronome to natural estuaries by the numerical model as $n_h = n_T = n_L$ for all settings, a constant $a$ for the tilting setup (dashed lines for tilt1 and dotted lines for tilt2) and a constant $a/h$ for the Reynolds method (drawn lines). Legend indicates length of the basin in m and relative downstream distance is from 0:landward to 1:inlet.

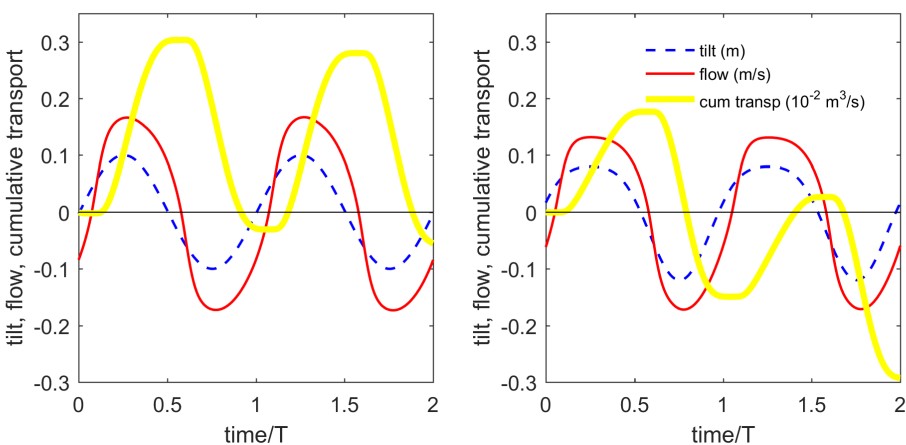

**Figure 17.** Flood-dominance accomplished in the tilting flume by adding an overtide to the tilting with an amplitude of 20% of the principal tide with a phase delay of $\pi/2$. Ebb flow is positive. The transport is cumulative in time to show the flood-dominance in two tidal cycles.