# Peer review of "Turning the tide: comparison of tidal flow by periodic sealevel fluctuation and by periodic bed tilting in scaled landscape experiments of estuaries"

_Earth Surface Dynamics, 2017_

## Referee Comment (RC1) · Anonymous Referee #1 · 5 May 2017

General comments

This paper is principally a methods paper. They describe an experimental apparatus, they put it through its paces, and they report the performance specs. This is an awkward place for a paper. If it is truly a methods paper, it provides inadequate information on how this system works. For example, their description for measuring flow velocity - THE primary variable to diagnose the success of their experiment - is incomplete and unclear. Moreover, this is not a "method" that is likely to be widely adopted since it is

a specific piece of hardware for a very specific purpose. Moreover, the authors have already published 3 papers(!) on the concept of a periodic tilting flume, albeit a smaller model, so the basic idea here is not new. This paper reports results of flows in a bigger version of an apparatus than they built before, and shows its superiority over the sea-level fluctuations. But, this was already shown in a way in the previous papers because they proved that tilting could move sand and make bars and shoals.

My personal take is that the authors should publish a paper when they have some science to show. Don't get me wrong, this is an impressive apparatus and it is likely to lead to new science insights. But in my opinion, experiments are not built to write papers about how they perform; they are built to learn new science. Lots of us have nice experimental equipment that we built. And if an experimental paper is written without science, it should at least be a very novel idea. The authors had a novel idea for sure, but that novelty has already been revealed in three previous papers. Based on this I believe this paper should be rejected - not because it is bad, but because it is incomplete. When the authors have some new significant science results - which they no doubt will in the future - they can write a nice paper and the current manuscript will serve as the methods section. But as is, this reads more like an internal laboratory technical report than a paper that is appropriate for ESurf. It may be appropriate for some engineering journal.

Detailed comments:

Abstract, line 5: ..."which in tidal systems with dynamic channel..." -> typo, or some words missing, in this sentence.

After a sentence saying that the "third, complementary method of research is controlled laboratory experiments", the authors cite the following paper: Kleinhans, M. G., van der Perk, M., and Bierkens, M.: On the use of laboratory experimentation: "Hydrologists, bring out shovels and garden hoses and hit the dirt", Hydrol. Earth Syst. Sci., 14, 369–382, 2010. There are so many papers laying out the rationale for geomorphology

**ESurfD**
experiments, that are more fundamental and much farther ranging than this one.

This problem about novelty is reinforced in the closing two sentences of the Discussion - the two most important points made about the utility of this setup each make reference to a previous Kleinhans et al. paper.

-> PIV involves strobing a light (often laser) and taking synced images of a desired exposure. This generates streaks in each image, whose length and orientation are used to produce a velocity vector field. It doesn't sound like what was done here. Particle Tracking Velocimetry (PTV) is another technique where trajectories of individual particles are traced by correlating them from image to image. I am guessing that this is what was done? Either way, there are no specifications about the resolution or error of the methods, and no demonsration that they resolve the flow correctly.

"ten images were collected at 25 Hz simultaneously by all cameras." -> So they only recorded for ∼0.3 seconds??

Note the following references that appear in the bibliography: Kleinhans, M. G., van der Vegt, M., Terwisscha van Scheltinga, R., Baar, A., and Markies, H.: Turning the tide: experimental creation of tidal channel networks and ebb deltas, Netherlands J. of Geoscience, 91, 311–323, 2012.

Kleinhans, M. G., van Rosmalen, T., Roosendaal, C., and van der Vegt, M.: Turning the tide: mutually evasive ebb- and flood-dominant channels and bars in an experimental estuary, Advances in Geosciences, 39, 21–26, doi:10.5194/adgeo-39-21-2014, 2014b.

Kleinhans, M. G., Terwisscha van Scheltinga, R., van der Vegt, M., and Markies, H.: Turning the tide: growth and dynamics of a tidal basin and inlet in experiments, J. of Geophys. Res. Earth Surface, 120, 95–119, doi:10.1002/2014JF003127, 2015.

Choosing articial grass as the channel boundary seems like a really strange and not very good idea. Why would you choose such an uncontrollable boundary - made of things that are bendy and make varied structures.
**ESurfD**

Interactive
comment

---

## Referee Comment (RC2) · Anonymous Referee #2 · 9 May 2017

This manuscript addresses a new type of the experimental facility for morphodynamics of tidal currents. The Metronome is an experimental flume to enable physical simulation of sediment transport processes by long-period oscillatory flows such as tidal currents. Judging from this preliminary report, the experimental technique employed in the Metronome can be evaluated with promising results. Thus, the topic addressed is interesting and deserves a constructive discussion in Earth Surface Dynamics Discussions. However, there are several issues to be clarified for publication.

Main issues are:

[Figure]

(1) The importance of the spatially non-uniform flows in developing estuarine geomorphology should be discussed in the paper. As the authors pointed out in Figure 1, one of the essential differences between the periodic tilting flume (Metronome) and natural estuaries is the spatial heterogeneity of flow discharges. In natural estuaries, flow height and velocity vary remarkably in space, whereas the Metronome produces quasi-uniform unsteady flows. Although this difference could be negligible for morphodynamic processes in estuaries, this should be discussed in detail at the discussion chapter.

(2) The purpose of comparison between experimental results and model calculation is unclear. To prove that setting of the facility is appropriate for experiments of estuaries, the experimental results should be compared with observation of natural estuaries or numerical simulations using conditions of the natural environments. However, the model calculation in this paper was conducted at the initial and boundary conditions similar to the experimental flume. Thus, the comparison can be validation of the model, not the experimental flume. It could be better to conduct numerical experiments with strongly non-uniform flow conditions, and to compare results with those of the periodic tilting flume.

(3) The descriptions of results often contain interpretation or speculative comments (ex. P.10 lines 1-8; P.11 lines 30-33; P.13 lines 15-19). These comments should be moved to the discussion chapter.

Minor comments:

(1) Definition of the parameter $Q$ is missing (probably flow discharge $Q = uwh$). In addition, the advection seems necessary in equation (2).

(2) Figure 12c shows that the model prediction of water level in the sand-bed channel experiments is very different from the measurements even from the qualitative viewpoints, but this figure is not cited and be addressed in the text.

**ESurfD**

Interactive
comment

---

## Author Comment (AC1) · 9 Jun 2017

We thank the reviewer for critical comments that help us to clarify the work. Minor suggestions will be followed or taken as an indication for the need of textual improvement.

The main comment is that the reviewer understood this to be a methods paper and therefore misses important elements in the methods and considers this journal the wrong outlet for this paper. We will make more clear in the future submission what the scientific issue is and analyse in more detail what aspects of tidal flow are reproduced

in the Metronome and why it goes wrong in the classic setup. We agree with the reviewer that the tilting principle is indeed not new and the morphology produced in small tilting flumes as published earlier is promising, but as a reviewer of one of those earlier papers rightfully remarked, there is no evidence that this morphology happens for the right reason. Moreover, it remained unclear why the classic experiments by Osborne Reynolds could not work even though he initiated tidal flow as in nature. So, whether tilting produces periodic flows that are sufficiently similar to tidal currents in nature for the purpose of landscape experiments is therefore still an open scientific question, and why the flows driven by periodic sealevel fluctuation, as in nature, are not sufficient for experiments is not fully clear. In the manuscript we target this question by direct flow measurement, allowing comparison to well-known properties of tidal flow, and by modelling, allowing identification of the most important terms in the physics describing tidal flow. This will benefit selection of proper settings in tidal flumes that are being built, it will answer the inevitable reiteration of the reviewer question about tidal flow in future submissions and therefore it will be cited in future papers. Our main result is that flow in the experiments is much more friction-dominated than in nature, meaning that the tidal wave generated at the seaward boundary dampens out too fast in experiments to cause sediment transport in the estuary, unlike the tilting setup where the pressure gradient is enforced along the estuary by the periodic tilting.

We also note that the problem of novelty perceived by the reviewer in relation to the closing sentences in the discussion is countered by Reviewer Comment 2: RC2 requires more, rather than less, elaboration of the possibilities of the present setup. This shows a need for brief discussion that couples the present paper to our earlier papers on small-scale experiments with tidal morphology. Moreover, for the resubmission we will conduct modelling at a range of scales from experiment to nature to explore at what scales the tilting setup ceases to perform well and the Reynolds setup begins to perform well.

The reviewer suggests we should cite different, more fundamental papers on the use

of laboratory experiments rather than our 2010 HESS paper, but provides no specific references. We chose this reference because it explains fundamental complementarity with modelling and field data analyses, all of which we use. This reference also refers to such more fundamental papers in earth science and in philosophy of science, but rehashing these in this paper as we did in our extensive ESR paper would take much space that we prefer to use for improved debate about the fundamental science questions. We will add some additional references though.

Here we explain the Particle Imaging Velocimetry (PIV) method better and we will add this to the paper. The method is the Large Scale Surface Particle Image Velocimetry (LS-PIV) developed by Uijttewaal et al. (2001) as also used in Blanckaert et al. (2012). This means that the particles are floating on the water surface and are lighted and repeatedly imaged from above as described in our manuscript. The image processing then proceeds as with classical PIV, in our case by peak cross-correlation, which is a windowing operation that does not identify individual particles as would be the case in Particle Tracking Velocimetry. Problems specific for the tilting flume are that the distance to the cameras changes over time, and that the flow conditions change rapidly. This led to the chosen measurement frequency, which is more than sufficient for data reduction to width-averaged flow velocity.

The choice of artificial soccer grass as the channel boundary needs more explanation and illustration. Two observations in past experiments led to this choice. First, channels get attracted to flume walls in many experiments, and perhaps also to valley walls in nature. This is due to two effects: first, the lower roughness of the sidewall compared to an alluviated bed with bedforms, and second, and lack of slope-driven sediment transport from a rocky sidewall sidewall that would potentially make a channel migrate towards the middle. The usual methods to prevent this are to use wider flumes and to add high banks of sediment, but even one locality of scour onto a smooth wall may self-amplify to strongly affect the large-scale morphology. Michal Tal and Chris Paola did an unpublished braided river experiment with groynes to try and push the channels out of the flume wall, but, as observed in nature, this caused local scours at each groyne that seemed to attract the flow rather than repel it. This is probably because the scale of the groyne spacing and scour holes approaches that of the channel width. The artificial grass, on the other hand, provides large roughness at a much smaller scale that prevents scour holes (Kleinhans et al. 2017). It is quite uniform and dense with rather rigid plastic stems that did not bend noticeably in the strongest experimental flows tested. As such, it also provided a large uniform roughness for the clear-water flow experiments reported in our manuscripts, it makes a gradual transition from alluviated sand-bed to a nearly fixated rough bed that is a more realistic bottom boundary than smooth steel because that would be much smoother than the sediment. Finally, it provides a soft protection for the thin steel floor of the flume. We therefore strongly recommend such artificial grass for landscape and bedform experimentation and will clarify this better in the manuscript.

Finally, we do not agree with the reviewer that papers with a methodological component do not fit in this journal. We consider this paper to have a strong methods component, but we target the science. At this moment, other papers considered for this journal report on rather methodological issues such as automated laser scanning for change detection, seismic signal analysis for rockfall dynamics measurement, measurements on geometry with an R-package, validation of digital elevation models, and so on and so forth. Clearly, there is a need to assess and publish whether and how these methods perform for specific and potentially more general subjects in the Earth Surface Dynamics and we believe this to be valid for our case as well. We think that the tilting flume is not a 'specific piece of hardware for a very specific purpose' as the reviewer suggests; rather, it is a technically simple but scientifically novel functionality added to a type of flume used worldwide for a great many different types of experiments of landscapes and geology, opening up new alleys for experimental biogeomorphology. Two new metronomes have already been built outside our institute for education and outreach purposes in the Netherlands and one metronome is being built in the United States for research on washovers. By providing information about the technical setup of

the Metronome we disseminate the idea such that others can use it, and the evidence in the present paper shows to what degree it simulates tidal flows.

References: Blanckaert, K., M.G. Kleinhans, S.J. McLelland, W.S.J. Uijttewaal, B.J. Murphy, A. van der Kruijs, D.R. Parsons and Q. Chen (2012). Flow separation at the inner (convex) and outer (concave) banks of constant-width and widening open-channel bends. Earth Surf. Proc. and Landforms 38, 696-716. Kleinhans, M.G., J. Leuven, L. Braat and A. Baar (2017). Scour holes and ripples occur below the hydraulic smooth to rough transition of movable beds. Sedimentology,DOI: 10.1111/sed.12358. Uijttewaal WSJ, Lehmann D, van Mazijk A. 2001. Exchange processes between a river and its groyne fields; model experiments. Journal of Hydraulic Engineering 127(11): 928–936.

---

## Author Comment (AC2) · 9 Jun 2017

We thank the reviewer for pointing out where the manuscript requires clarification. Minor suggestions will be followed or taken as an indication for the need of textual improvement.

(Comment 1) Discussing spatially non-uniform flows is an interesting suggestion that we will follow up in resubmission. We think two aspects relevant for morphology require such discussion. The simplest aspect of spatial non-uniformity is that the phase

of the tidal flow varies along natural estuaries but is enforced to be simultaneous in the Metronome. In order for this to have morphological effects, a the velocity gradient resulting from a phase difference would need to be so strong that it has a morphological effect. Some bar theory however proceeds from the assumption of rigid lid, suggesting no importance of phase differences for morphology. Furthermore, bar morphology develops over a great many tidal cycles for which the precise timing of the tide is not important. However, we will analyse phase differences in more detail in the resubmission because the data and model show that, even in the simplest possible conditions of a straight uniform tidal channel, the Metronome shows nonuniformity when the upstream boundary is closed and this requires brief discussion of its potential effects on morphology.

The second, more complicated difference arises at the local scale of parallel channels around bars where phase differences between the horizontal tide and the vertical tide may emerge. In natural systems the tidal wave propagation depends on the channel depth. This leads to time-varying water level differences between parallel channels that drive currents across the bars separating the channels. How this affects bar morphology, dynamics and formation of tie channels and new channel bifurcations is an open question (Swinkels et al.2009). It is known from braided rivers that water level differences are caused by backwater effects due to non-uniform depth along channels have significant effects on the braiding processes (Schuurman & Kleinhans 2015), so the expectation is that additional water level differences between tidal channels will also have significant effects on morphology. So the question is whether such tidal phase differences can arise in the Metronome given the uniformly prescribed tidal phase. One of us conducted measurements on water level and flow velocity around self-formed tidal bars in the Metronome and processing will determine how this works and compare to full-scale flows in numerical modelling to unravel effects of such phase differences. We will pay attention to this issue in future submissions on experiments with freely developing morphology, that is outside the scope of the present paper.
(Comment 2) The purpose of the comparison between the measurements and the modelling is not clear according to the reviewer, because no natural conditions are modelled, suggesting that the measurements serve to validate the modelling. This gave us the idea to use the model in the resubmission to investigate at what length scales tilting or Reynolds-type experiments begin to give realistic results and we intend to apply the model at a range of scales from a mini-metronome to the full scale of natural systems in the resubmission. At what scales tilting or Reynolds-type experiments produce periodic flows that are sufficiently similar to tidal currents in nature for the purpose of landscape experiments is still an open question. In the manuscript we targeted this question by direct flow measurement, allowing comparison to well-known properties of tidal flow, and by modelling, allowing identification of the most important terms in the physics describing tidal flow. The fact that the model is the same as often used to simulate full-scale tidal flows indicates that the relevant physical mechanisms are the same as in natural tides.

The reviewer suggests that we should have used the model for strongly non-uniform flow conditions to compare against the flows in the periodic tilting flume. We do not understand what the reviewer means by strongly non-uniform other than having bars and bends, which is beyond the scope of this paper. However, we agree that comparison between conventionally driven tidal flow is needed. We need to stress more in the manuscript that this is exactly what we did by numerical modelling of the Reynolds setup at the scale of the flume. We will also apply the model at the natural scale and analyse water level and velocity amplitudes, phase differences and sediment mobility.

(Comment 3) The indicated sentences with interpretation were originally placed in the results section for reasons of readability. We will move them to the discussion in the next manuscript version. (minor comment 1) We will add the definition of Q. Before submission we debated whether advection is indeed necessary in equation 2 and came to the conclusion that it is not, for the question at hand. For example, the advection term is about an order of magnitude smaller than the advection term. We will analyse

and discuss this in the resubmission.

(minor comment 2) We thank the reviewer for identifying this omission. The phase of the modelled water levels is indeed rather different from the observed water levels, but the magnitude is the same as measured except near the transition from the sea to the tidal channel. The reason is that the 1D model cannot reproduce a 2D phenomenon, namely the extreme and sudden convergence from the sea into the channel. In these sand-bed experiments the morphology is more realistic in the sense of roughness and channel dimensions, but less realistic in the sense of this artificial transition that arises because we chose to do the simplest idealised scenario rather than a more realistic convergent shape that is more difficult to interpret. This, in hindcast, led to qualitatively similar but quantitatively different model results.

References:

Schuurman, F. and M.G. Kleinhans (2015). Bar dynamics and bifurcation evolution in a modelled braided sand-bed river. Earth Surf. Proc. and Landforms 40, 1318-1333

Swinkels, C.M., Jeuken, C.M.,Wang, Z.B., Nicholls, R.J. (2009). Presence of connecting channels in the Western Scheldt Estuary. J. Coast. Res. 627–640.

---

## Author Response (AR1)

Utrecht, 6 July 2017

To the editor,

Below we detail how we addressed reviewer comments. These are mostly the same as the reviewer replies posted online in the discussion phase.

Sincerely, Maarten, on behalf of all authors

**RC1**

We thank the reviewer for critical comments that help us to clarify the work. Minor suggestions will be followed or taken as an indication for the need of textual improvement.

The main comment is that the reviewer understood this to be a methods paper and therefore misses important elements in the methods and considers this journal the wrong outlet for this paper. We will now make more what the scientific issue is and analyse in more detail what aspects of tidal flow are reproduced in the Metronome and why it goes wrong in the classic setup. We agree with the reviewer that the tilting principle is indeed not new and the morphology produced in small tilting flumes as published earlier is promising, but as a reviewer of one of those earlier papers rightfully remarked, there is no evidence that this morphology happens for the right reason. Moreover, it remained unclear why the classic experiments by Osborne Reynolds could not work even though he initiated tidal flow as in nature. So, whether tilting produces periodic flows that are sufficiently similar to tidal currents in nature for the purpose of landscape experiments is therefore still an open scientific question, and why the flows driven by periodic sealevel fluctuation, as in nature, are not sufficient for experiments is not fully clear. In the manuscript we target this question by direct flow measurement, allowing comparison to well-known properties of tidal flow, and by modelling, allowing identification of the most important terms in the physics describing tidal flow. This will benefit selection of proper settings in our flume and in tidal flumes that are being built, it will answer the inevitable reiteration of the reviewer question about tidal flow in future submissions. Our main result is that flow in the experiments is much more friction-dominated than in nature, meaning that the tidal wave generated at the seaward boundary dampens out too fast in experiments to cause sediment transport in the estuary, unlike the tilting setup where the pressure gradient is enforced everywhere along the estuary by the periodic tilting.

We also note that the problem of novelty perceived by the reviewer in relation to the closing sentences in the discussion is countered by Reviewer Comment 2: RC2 requires more, rather than less, elaboration of the possibilities of the present setup. This shows a need for brief discussion that couples the present paper to our earlier papers on small-scale experiments with tidal morphology. Moreover, we conducted modelling at a range of scales from experiment to nature to explore at what scales the tilting setup ceases to perform well and the Reynolds setup begins to perform well. Also we show by modelling that tidal asymmetry can be enforced by tilting with an overtide, so that flood-dominant conditions can be obtained for the purpose in estuary infilling.

The reviewer suggests we should cite different, more fundamental papers on the use of laboratory experiments rather than our 2010 HESS paper, but provides no specific references. We chose this reference because it explains fundamental complementarity with modelling and field data analyses, all of which we use. This reference also refers to such more fundamental papers in earth science and in philosophy of science. We added some additional references but chose not to rehash fundamental papers in the present manuscript but refer to our extensive ESR paper where we discussed these matters in detail.

We now explain the Particle Imaging Velocimetry (PIV) method better and added references. The method is the Large Scale Surface Particle Image Velocimetry (LS-PIV) developed by Wim Uijttewaal as used in Blanckaert et al. (2012) and Marra et al. 2014. This means that the particles are floating on the water surface and are lighted and repeatedly imaged from above as described in our manuscript. The image processing then proceeds as with classical PIV, in our case by peak cross-correlation, which is a windowing operation that does not identify individual particles as would be the case in Particle Tracking Velocimetry. Problems specific for the tilting flume are that the distances to the cameras change over time, and that the flow conditions change rapidly. This led to the chosen measurement frequency and duration, which is more than sufficient for data reduction to width-averaged flow velocity. Our ongoing experiments (not reported here) show that this is even sufficient to collect 2D flow fields.

The choice of artificial soccer grass as the channel boundary is now explained better. Two observations in past experiments led to this choice. First, channels get attracted to flume walls in many experiments, and perhaps also to valley walls in nature. This is due to two effects: first, the lower roughness of the sidewall compared to an alluviated bed with bedforms, and second, and lack of slope-driven sediment transport from a rocky sidewall sidewall that would potentially make a channel migrate towards the middle. The usual methods to prevent this are to use wider flumes and to add high banks of sediment, but even one locality of scour onto a smooth wall may self-amplify to strongly affect the large-scale morphology. Michal Tal and Chris Paola did an unpublished braided river experiment with groynes to try and push the channels out of the flume wall, but, as observed in nature, this caused local scours at each groyne that seemed to attract the flow rather than repel it. This is probably because the scale of the groyne spacing and scour holes approaches that of the channel width. The artificial grass, on the other hand, provides large roughness at a much smaller scale that prevents scour holes (Kleinhans et al. 2017). It is quite uniform and dense with rather rigid plastic stems that did not bend noticeably in the strongest experimental flows tested. As such, it also provided a large uniform roughness for the clear-water flow experiments reported in our manuscripts, it makes a gradual transition from alluviated sand-bed to a nearly fixated rough bed, when the bed is scoured to that depth, that is a more realistic bottom boundary than smooth steel because that would be much smoother than the sediment. Finally, it provides a soft protection against shovels for the thin steel floor of the flume. We therefore strongly recommend such artificial grass for landscape and bedform experimentation and clarified this better in the manuscript.

Finally, we do not agree with the reviewer that papers with a methodological component do not fit in this journal. We consider this paper to have a strong methods component, but we target the science. At this moment, other papers considered for this journal report on rather methodological issues such as

automated laser scanning for change detection, seismic signal analysis for rockfall dynamics measurement, measurements on geometry with an R-package, validation of digital elevation models, and so on and so forth. Clearly, there is a need to assess and publish whether and how these methods perform for specific and potentially more general subjects in the Earth Surface Dynamics and we believe this to be valid for our case as well. We think that the tilting flume is not a 'specific piece of hardware for a very specific purpose' as the reviewer suggests; rather, it is a technically simple but scientifically novel functionality added to the sandbox type of flume used worldwide for a great many different types of experiments of landscapes and geology, opening up new alleys for experimental biogeomorphology. Indeed, two new metronomes have already been built outside our institute for education and outreach purposes in the Netherlands and one metronome is being built in the United States for research on washovers. By providing information about the technical setup of the Metronome we disseminate the idea such that others can use it, and the evidence in the present paper shows to what degree it simulates tidal flows.

**References**

Blanckaert, K., M.G. Kleinhans, S.J. McLelland, W.S.J. Uijttewaal, B.J. Murphy, A. van der Kruijs, D.R. Parsons and Q. Chen (2012). Flow separation at the inner (convex) and outer (concave) banks of constant-width and widening open-channel bends. Earth Surf. Proc. and Landforms 38, 696-716.

Kleinhans, M.G., J. Leuven, L. Braat and A. Baar (2017). Scour holes and ripples occur below the hydraulic smooth to rough transition of movable beds. Sedimentology, DOI: 10.1111/sed.12358.

Marra, W. A., Parsons, D. R., Kleinhans, M. G., Keevil, G.M., and Thomas, R. E.: Near-bed and surface flow division patterns in experimental river bifurcations, Water Resources Research, 50, 1506–1530, doi:10.1002/2013WR014215

2014.

**RC2**

We thank the reviewer for pointing out where the manuscript requires clarification. Minor suggestions were followed or taken as an indication for the need of textual improvement.

(Comment 1) Discussing spatially non-uniform flows is an interesting suggestion that we now discuss. We think two aspects relevant for morphology require such discussion. The simplest aspect of spatial non-uniformity is that the phase of the tidal flow varies along natural estuaries but is enforced to be simultaneous in the Metronome. In order for this to have morphological effects, a the velocity gradient resulting from a phase difference would need to be so strong that it has a morphological effect. Some bar theory however proceeds from the assumption of rigid lid, suggesting no importance of phase differences for morphology. Furthermore, bar morphology develops over a great many tidal cycles for which the precise timing of the tide is not important, a point now stressed in the introduction. However, even in the simplest possible conditions of a straight uniform tidal channel, the Metronome shows nonuniformity when the upstream boundary is closed and this led to brief discussion of its potential effects on morphology.

The second, more complicated difference arises at the local scale of parallel channels around bars where phase differences between the horizontal tide and the vertical tide may emerge. In natural systems the tidal wave propagation depends on the channel depth. This leads to time-varying water level differences between parallel channels that drive currents across the bars separating the channels. How this affects bar morphology, dynamics and formation of tie channels and new channel bifurcations is an open question (Swinkels et al.2009). It is known from braided rivers that water level differences are caused by backwater effects due to non-uniform depth along channels have significant effects on the braiding processes (Schuurman & Kleinhans 2015), so the expectation is that additional water level differences between tidal channels will also have significant effects on morphology. So the question is whether such tidal phase differences can arise in the Metronome given the uniformly prescribed tidal phase. One of us conducted measurements on water level and flow velocity around self-formed tidal bars in the Metronome and processing will determine how this works and compare to full-scale flows in numerical modelling to unravel effects of such phase differences. We will pay attention to this issue in future submissions on experiments with freely developing morphology, that is outside the scope of the present paper.

(Comment 2) The purpose of the comparison between the measurements and the modelling was not clear according to the reviewer, because no natural conditions are modelled, suggesting that the measurements serve to validate the modelling. This stimulated us to use the model in the resubmission to investigate at what length scales tilting or Reynolds-type experiments begin to give realistic results. We already had the idea to apply the model at a range of scales from a mini-metronome to the full scale of natural systems but not yet for this paper, but now we do so in this resubmission. At what scales tilting or Reynolds-type experiments produce periodic flows that are sufficiently similar to tidal currents in nature for the purpose of landscape experiments is still an open question. In the manuscript we targeted this question by direct flow measurement, allowing comparison to well-known properties of tidal flow, and by modelling, allowing identification of the most important terms in the physics describing tidal flow for the scaling of the estuary and the potential morphodynamics.

The reviewer suggests that we should have used the model for strongly non-uniform flow conditions to compare against the flows in the periodic tilting flume. We do not understand what the reviewer means by strongly non-uniform other than having bars and bends, which is beyond the scope of this paper. However, we agree that comparison between conventionally driven tidal flow is needed. We need to stress more in the manuscript that this is exactly what we did by numerical modelling of the Reynolds setup at the scale of the flume. We also applied the model at the natural scale and analysed the resulting water level and velocity amplitudes, phase differences and sediment mobility as well as tidal excursion lengths and tidal wavelengths

(Comment 3) The indicated sentences with interpretation were originally placed in the results section for reasons of readability. We moved them to the discussion.

(minor comment 1) We added the definition of Q. Before submission we debated whether advection is indeed necessary in equation 2 and came to the conclusion that it is not, for the question at hand. The

advection term is about an order of magnitude smaller than the pressure and friction terms. We analysed this and discuss it in the resubmission.

(minor comment 2) We thank the reviewer for identifying this omission. The phase of the modelled water levels is indeed rather different from the observed water levels, but the magnitude is the same as measured except near the transition from the sea to the tidal channel. The reason is that the 1D model cannot reproduce a 2D phenomenon, namely the extreme and sudden convergence from the sea into the channel. In these sand-bed experiments the morphology is more realistic in the sense of roughness and channel dimensions, but less realistic in the sense of this artificial transition that arises because we chose to do the simplest idealised scenario rather than a more realistic convergent shape that is more difficult to interpret. This, in hindcast, led to qualitatively similar but quantitatively different model results.

```
\documentclass[esurf, manuscript]{copernicus}

%\documentclass[esurf]{copernicus}

\usepackage{rotating}

%\graphicspath{{figures/}}

\graphicspath{{D:/Wordfiles/Metronome/Flowmeasurements/}}

%\graphicspath{{C:/Users/Maarten
Kleinhans/Documents/Wordfiles/Metronome/Flowmeasurements}}

\begin{document}

\title{Turning the tide: comparison of tidal flow by periodic sealevel fluctuation and by periodic bed
tilting in the Metronome tidal facilityscaled landscape experiments of estuaries}

\Author[1]{Maarten G.}{Kleinhans}

\Author[1]{Maarten}{van der Vegt}

\Author[1]{Jasper}{Leuven}

\Author[1]{Lisanne}{Braat}

\Author[1]{Henk}{Markies}

\Author[2]{Arjan}{Simmelink}

\Author[1]{Chris}{Roosendaal}

\Author[1]{Arjan}{van Eijk}

\Author[1]{Paul}{Vrijbergen}

\Author[1]{Marcel}{van Maarseveen}

\affil[1]{Faculty of Geosciences, Utrecht University, PObox 80115, 3508 TC Utrecht, The Netherlands}
```

\affil[2]{Consmema design and steel construction, Hattem, The Netherlands}

\runningtitle{Flow in tidal system experiments}

\runningauthor{Kleinhans et al.}

\correspondence{M.G. Kleinhans, m.g.kleinhans@uu.nl}

\received{}

\pubdiscuss{} %% only important for two-stage journals

\revised{}

\accepted{}

\published{}

\firstpage{1}

\maketitle

\begin{abstract}

Analogue models or scale experiments of estuaries and short tidal basins are notoriously difficult to create in the laboratory because of the difficulty to obtain currents strong enough to transport sand. Since Osborne Reynolds' experiments over a century ago, experimental tidal flow has been driven by periodic sealevel fluctuations. Recently we discovered a novel method to drive periodic tidal currents:  periodically tilting the entire flume. This leads to sediment transport in both the ebb and flood phase which in tidal systems withcausing dynamic channel and shoal patterns. However, it remains unclear whether tilting produces periodic flows that are sufficiently similar to tidal currents in nature for the purpose of landscape experiments. Moreover, it is not well understood why the flows driven by periodic sealevel fluctuation, as in nature, are not sufficient for morphodynamic experiments. Here we compare the tidal currents driven by sealevel fluctuations with those driven by tilting. We use in experiments, and scale these results up by a bespoke flume of 20 m by 3 m with rough bed: the Metronome.numerical model to compare flow and sediment mobility with large natural estuaries. Experiments were run in a 20 by 3 m straight flume with different, the Metronome, for a range of tilting periods and with either one or bothtwo boundaries open at constant head with free inflow and outflow. Also experiments were run with flow driven by periodic sealevel

fluctuations. We recorded surface flow velocity along the flume with Particle Imaging Velocimetry and water levels along the flume. We compared the results to a one-dimensional model with shallow flow equations for a rough bed, which was tested on the experiments and applied to a range of length scales bridging small experiments and large estuaries. We found that Reynolds' method results in negligible flows along the flume except for the first few meters, whereas flume tilting results in nearly uniform, reversing flow velocities along the entire flume that are strong enough to move sand. Furthermore, tidal excursion length relative to basin length is similar in tidal experiments and reality. Where Reynolds' method is limited by small sediment mobility and high tidal range relative to water depth, the tilting method allows independent control over the variables flow depth, velocity, sediment mobility, tidal period and excursion length, and tidal asymmetry. A periodically tilting flume thus opens up the possibility of systematic biogeomorphological experimentation with self-formed estuaries.

\end{abstract}

%novelty, originality and importance: This paper reports on a new experimental facility that allows live-bed landscape experiments of tidal systems. Although some experiments in a much smaller pilot setup are published, this much larger facility is presently used for experiments similar to the breakthrough braided and meandering river experiments in several labs, only now with tidal flow. This paper provides experimental and numerical evidence that the setup by Osborne Reynolds, used for over a century, is flawed whilst the novel setup works to create flows with mobile sediment. Simulations with a bespoke numerical model show that upscaling from experiment to natural systems requires vertical distortion, while characteristic relative length scales of tidal wavelength and tidal excursion length remain reasonably close to those of natural systems. Moreover, the tilting allows independent control of these variables, sediment mobility and tidal asymmetry.

%reviewer suggestions: a.l.densmore@durham.ac.uk, cpaola@umn.edu, H.H.G.Savenije@tudelft.nl, stijn.temmerman@uantwerpen.be, Wonsuck delta@jsg.utexas.edu

\introduction  %% \introduction[modified heading if necessary]

[revised manuscript text omitted]

\begin{acknowledgements}

Comments by two anonymous reviewers and guidancediscussion by the editors will beJob Dronkers are gratefully acknowledged. The research by MK, JL and LB and the construction of the Metronome were funded as a Vici grant to MK by the Dutch Technology Foundation (STW) that is part of the Netherlands Organisation for Scientific Research (NWO, grant 016.140.316). We acknowledge collaboration with Variodrive, who designed and programmed the actuators, Bart Boshuizen (TU-Delft) who programmed the motion, HiH engineering, who modelled the forces on the steel construction, the actuators and the floor, and Stemmer Imaging, who designed and programmed the imaging system. Data and building plans of the Metronome and of a 3~m long steel mini-Metronome available from MK upon request. \end{acknowledgements}

Data and building plans of the Metronome and of a 3~m long steel mini-Metronome, original images, Matlab image processing scripts and the Matlab flow model are available upon request from MK.

\end{acknowledgements}

\bibliographystyle{copernicus}

\bibliography{kleinhans,rivers,coast}

%% URLs and DOIs can be entered in your BibTeX file as:

%% URL = {http://www.xyz.org/~jones/idx_g.htm}

%% DOI = {10.5194/xyz}

\clearpage

%%% TABLES

\begin{table*}[t]

\caption{Boundary conditions applied in all experiments: auxiliary fixed-slope experiments to determine the roughness of the artificial grass bed, periodic tilting experiments with one or two open boundaries, and periodic sealevel variations. Experiments with sand bed were conducted with a shallow sea of 2~m length to reduce boundary effects and were closed on the upstream boundary.}

%The same conditions apply to the straight flume section and the convergent section.}

\begin{tabular}{lcccll}

\tophline

bed & period & tilt slope amplitude & sealevel amplitude & boundaries & rationale \\

\unit{} & \unit{s} & $\times10{^-3}$\unit{m/m} & $\times10{^-3}$\unit{m} & \unit{} & \unit{} \\

\middlehline

grass & & 0.9 & 0 & both open & steady flow: control \\

grass & & 2.3 & 0 & both open & same, faster flow \\

grass & 60 & 0 & 10 & x=0~m closed & same, longer wave \\

grass & 30 & 0 & 20 & x=0~m closed & Reynolds method \\

grass & 30 & 9.1 & 0 & both open & reach within estuary \\

grass & 15 & 9.1 & 0 & both open & same, shorter tidal excursion length \\

grass & 30 & 4.5 & 0 & both open & reduced tidal energy (not shown) \\

grass & 30 & 9.1 & 0 & x=0~m closed & basin with reflective landward boundary \\

grass & 15 & 9.1 & 0 & x=0~m closed & same, short tidal excursion length \\

sand & 40 & 3.6 & 0 & x=0~m closed & tilting, natural roughness \\

sand & 40 & 0     & 3.5 & x=0~m closed & Reynolds method, natural roughness \\

\bottomhline

\end{tabular}

\belowtable{} % Table Footnotes

\label{tab:expersettings}

\end{table*}

%\begin{table*}[h]

\begin{sidewaystable}

\caption{Properties of Reynolds and tilting estuaries across a range of scales from experiments to natural estuaries. Linear friction is defined as $r=8gu/(3\pi hC^2)$ with the given velocity rather than the unity value that is usually assumed. The ratio between friction and inertia is calculated as $rT/2\pi$. Velocities for the model scenarios were taken from the model runs at relative length 0.8. Experiments with the Reynolds setup were taken from \cite{reynolds1889} and \citet{tambroni2005} and with the tilting setup from \citet{kleinhans2014}. Data of the Dovey (UK) are from \citet{brown2010}, Thames from \citet{friedrichs2010} and Westerschelde from \citet{wang2002}.}

\begin{tabular}{rllllll|llll}

\tophline

case & Metronome & Kleinhans & Fig. 2 & Metronome & Tambroni 1 & Reynolds tank A & prototype & Dovey & Thames & Westerschelde \\

 & model & experiment & experiment & model & experiment & experiment & model & nature & nature & nature \\

configuration & tilt & tilt2 & tilt1 & Reynolds & Reynolds & Reynolds & Reynolds & Reynolds & Reynolds & Reynolds \\

\middlehline

```latex
length $L$ (m) & 20 & 3.5 & 20 & 20 & 24.14 & 3.62 & 20000 & 20000 & 95000 & 200000 \\

width $W$ (m) & 1.5 & 1.3 & 1.5 & 1.5 & 0.3 & 1.18 & 1500 & 800 & 4300 & 6000 \\

depth $h$ (m) & 0.03 & 0.004 & 0.025 & 0.03 & 0.082 & 0.05 & 30 & 5 & 8.5 & 15 \\

amplitude $a$ (m) & 0.1 & 0.018 & 0.05 & 0.01 & 0.05 & 0.05 & 10 & 2 & 2 & 1.75 \\

period $T$ (s) & 40 & 72 & 30 & 40 & 180 & 53 & 40000 & 44712 & 44712 & 44712 \\

period $T$ (h) &  &  &  &  &  & & 11 & 12.42 & 12.42 & 12.42 \\

Ch\'{e}zy (m$^{0.5}$/$s) & 11 & 11 & 11 & 11 & 11 & 11 & 35 & 35 & 50 & 55 \\

\middlehline

$u$ (m/s) & 0.2 & 0.1 & 0.25 & 0.07 & 0.24 &  & 1 & 1.2 & 1 & 1.5 \\

$\theta$ (-) & 1 & 0.25 & 0.5 & 0.11 & 0.33 &  & 10 & 1.99 & 1.26 & 2.56 \\

$Fr$ (-) & 0.37 & 0.5 & 0.5 & 0.13 & 0.27 &  & 0.06 & 0.17 & 0.11 & 0.12 \\

\middlehline

excursion $L_e$ (m) & 4 & 3.6 & 3.75 & 1.4 & 21.6 &  & 20000 & 26827 & 22356 & 33534 \\

wavelength $L_t$ (m) & 22 & 14 & 15 & 22 & 161 & 37 & 686000 & 313000 & 408000 & 542000 \\

friction $r$ (-) & 0.46 & 1.72 & 0.69 & 0.16 & 0.2 &  & 0.0002 & 0.0016 & 0.00039 & 0.00028 \\

\middlehline

 aspect $W/h$ & 50 & 325 & 60 & 50 & 4 & 24 & 50 & 160 & 506 & 400 \\

amplitude $a/h$ &  &  &  & 0.33 & 0.61 & 1 & 0.33 & 0.4 & 0.24 & 0.12 \\

 excursion $L_e/L$ & 0.2 & 1.03 & 0.19 & 0.07 & 0.89 &  & 1 & 1.34 & 0.24 & 0.17 \\

wavelength $L_t/L$ & 1.1 & 4 & 0.8 & 1.1 & 6.7 & 10.2 & 34.3 & 15.7 & 4.3 & 2.7 \\

$L_t/L_e$ & 5.5 & 3.9 & 4 & 15.7 & 7.5 &  & 34 & 12 & 18 & 16 \\

friction / inertia & 2.9 & 19.7 & 3.3 & 1 & 5.7 &  & 1.3 & 11.4 & 2.8 & 2 \\

\bottomhline

\end{tabular}

\label{tab:scalenumbers}

%\end{table*}
```

\end{sidewaystable}

\clearpage

%% FIGURES

\begin{figure*}[t]

\includegraphics[width=186cm]{principle.png}

[revised manuscript text omitted]

\label{fig:asymmetrictilt}

\end{figure*}

\end{document}

---

## Author Response (AR2)

To the editor,

Below we respond to reviewer 2 and steer by the AE in *italics* text. Reviewer 1 had no comments. This document also contains the tracked changes in the LaTeX file. In addition to the changes in response to the review there are also a few minor changes in phrasing.

Furthermore a paragraph with hypothetical explanations for the differences between the experiments and modelling was taken out of the discussion and put as a separate subsection at the end of the relevant results section with a header clearly indicating that this was interpretation rather than result. We realise that in the previous review round the reviewers objected against interpretation in the results and hope that the clearer formulation and separate subheading flags this sufficiently. In the extended discussion this paragraph was out of place in that it did not fit in the flow and was much more specific to certain detailed results that were not really relevant to the bigger story. If the editor prefers different we will put it back.

Sincerely, Maarten, on behalf of all authors

**Associate Editor Decision: Publish subject to minor revisions Daniel Parsons**

The authors have made some significant changes in light of the first review. These address the majority of the comments. However, there remain a few, still substantive, issues that need to be addressed via some minor to moderate revision prior to publication ....doing so will ensure that the paper has a broader reach.

*We thank the Associate Editor for his steer. Changes indicated below are also reflected in the abstract and conclusions.*

In essence the issue centres on the fact that this paper still reads in most part like a methods paper - which in itself is not a problem, as Esurf accepts and publishes methods-based papers. However, methods papers need to be novel and this is not as novel as it could be - a smaller scale Metronome paper already exists that explains the majority of the background physics and theory. To counter this, the authors include some primary science elements - which are welcomed - but these would typically require some novel scientific conclusions. But, these read, at present, very secondary to the methodological elements of the paper. As such the paper occupies some uncomfortable middle ground.

*We now more clearly state that the novelty lies in the first **comparison** ever between the Reynolds method and the tilting method in terms of flow conditions, sediment mobility and typical dimensionless scales for tidal behaviour. This was not done in our previous work except by qualitative reasoning. In fact one reviewer and the AE on our JGR 2015 paper remarked that they really wanted to see such a comparison. However, at that time and in that small facility we could not measure the flow in the required detail. Furthermore our earlier work focussed on tidal inlet systems with simple hydrodynamics, not estuaries with more complicated tidal behaviour as we already state in the introduction.*

To address this I suggest that the authors refocus some of the text to address the below two remaining points:

1. Highlight the scale of the facility compared to the smaller test facility and detail how the scale means that the large facility is able to capture qualitatively new behaviours from the smaller facility? i..e what is the novel aspects of behaviour that can be modelled by making the facility bigger?

*We now devote some discussion to this. There is a major difference in the friction dominance over inertia and also in the mobility and Froude number. We added these numbers to the table and discuss them. Most importantly, however, the size will make a difference in matters not presented here but now briefly explained on the basis of our earlier experiments with rivers: the formation of floodplain by mud simulants and vegetation is much more difficult to get right in the smaller facility because the turbulence drops out of the flow above bars much sooner than in larger facilities and because vascular plants just don't come in smaller species on this planet. The practical point that measurements of patterns and processes are much easier in a larger flume is also briefly mentioned.*

2. In driving more towards an enhanced methods paper I would also like to see some additional detail on the methods included. At present there is not sufficient detail so that others could re-produce the experiments. Including these additional details (even as supplementary) would ensure that the paper would be used and cited a greater number of times.

*It is not entirely clear which methods this refers to. The PIV is standard and has been published by others in a number of papers. The method of tilting is a technical matter and it makes no difference for the flow whether one tilts the flume by a servo with excenter or by sophisticated actuators.*

Addressing these two points constitute minor to moderate revisions and I recommend that the paper be accepted once these have been attended to and I believe a final editor-only review will be sufficient.

**Reviewer 2**

This paper has been revised, with some improvements. The principle one is the use of the model to do a scaling up analysis of lab to field - which helps to understand what the Metronome can and cannot say about natural estuaries. That said, the authors did not address my major concerns. A few important points here:

1. I do not object to a Methods paper, and am fully aware that ESurf publishes Methods papers.

2. Any paper should be Novel.

The authors spill a fair bit of ink countering my objections in their rebuttal letter but they misrepresent them. They indicate that I think ESurf shouldn't publish Methods papers, and then they address the novelty issue in isolation; but the two are related. My main point is that, if it is a Methods paper, the method should be novel. And, in this case, it isn't: a smaller scale Metronome already exists and has been published on. So then, if it's a Science paper, there should be novel scientific conclusions. But,

there aren't really - although the situation has improved a bit with this version. So this paper STILL occupies an uncomfortable middle ground.

*See response to AE. We already stated that the novelty lies in a quantitative comparison of flow in both types of setups. This has to our knowledge been presented nowhere else in literature and is both novel and useful, which is why we did it in the first place. We now state more clearly that this is novel.*

Perhaps I need to be more direct here so that I am not misunderstood.

*Thank you.*

1. This is a bigger version of a previous experiment. Does it capture qualitatively new behaviors from the smaller one? Or, put another way, why make it bigger?

*See response to AE: for practical reasons and for scaling reasons, in particular of the floodplain processes.*

2. (related) There is a scaling up comparison from the Metronome to the field. How about from the Metronome to the previous smaller experiment?

*That is already in the paper by inclusion of that smaller scale in both the new figure and the table. We now discuss this more extensively.*

3. I still think there is a little too much self citation here, but I accept that this is subjective. I do not wish to go through case by case and suggest specific other papers; if the authors believe that theirs are the best, fine.

*We feel there is no need to clutter the paper with a large number of excellent scaling papers for river experiments when these have already been synthesised in recent review papers and the present focus is on tidal systems. Also it can be expected that the most applicable scaling considerations for our research interests are found in our earlier work. Unfortunately the reviewer did not provide suggestions.*

4. The intro states that the experiment is described so others could reproduce it. There has been little done since the previous version to describe the methods more fully, and access by others to data and materials associated with this paper is to be done by request from one of the authors. Again, this seems not too accessible as a Methods paper.

*It is not clear to us which methods the reviewer refers to. The PIV code is online already and our code did nothing but read out our images and use that existing code. We added the source website and some more details on the precise settings. The description of the tilting basin is such that anyone can reproduce it and this is what we intended by the remark in the intro and we attempted a better formulation now. (In fact, one handyman, two teachers and a group of highschool children made it, because the only thing that matters is that a box with sand and water can tilt periodically.) It is unfortunately not clear where we misunderstand the reviewer.*

```
\documentclass[esurf, manuscript]{copernicus}
%\documentclass[esurf]{copernicus}

\usepackage{rotating}

%\graphicspath{{figures/}}
\graphicspath{{D:/Wordfiles/Metronome/Flowmeasurements/}}
%\graphicspath{{C:/Users/Maarten Kleinhans/Documents/Wordfiles/Metronome/Flowmeasurements}}

\begin{document}

\title{Turning the tide: comparison of tidal flow by periodic sealevel fluctuation and by periodic bed
tilting in scaled landscape experiments of estuaries}

\Author[1]{Maarten G.}{Kleinhans}
\Author[1]{Maarten}{van der Vegt}
\Author[1]{Jasper}{Leuven}
\Author[1]{Lisanne}{Braat}
\Author[1]{Henk}{Markies}
\Author[2]{Arjan}{Simmelink}
\Author[1]{Chris}{Roosendaal}
\Author[1]{Arjan}{van Eijk}
\Author[1]{Paul}{Vrijbergen}
\Author[1]{Marcel}{van Maarseveen}

\affil[1]{Faculty of Geosciences, Utrecht University, PObox 80115, 3508 TC Utrecht, The Netherlands}
\affil[2]{Consmema design and steel construction, Hattem, The Netherlands}

\runningtitle{Flow in tidal system experiments}
\runningauthor{Kleinhans et al.}
\correspondence{M.G. Kleinhans, m.g.kleinhans@uu.nl}

\received{}
\pubdiscuss{} %% only important for two-stage journals
\revised{}
\accepted{}
\published{}

\firstpage{1}

\maketitle

\begin{abstract}
```

Analogue models or scale experiments of estuaries and short tidal basins are notoriously difficult to
create in the laboratory because of the difficulty to obtain currents strong enough to transport sand.

Our recently discovered  method to drive  tidal currents by periodically tilting the entire flume leads to intense sediment transport in both the ebb and flood phase causing dynamic channel and shoal patterns. However, it remains unclear whether tilting produces periodic flows with characteristic tidal properties that are sufficiently similar to those in nature for the purpose of landscape experiments. Moreover, it is not well understood why the flows driven by periodic sealevel fluctuation, as in nature, are not sufficient for morphodynamic experiments. Here we compare for the first time the tidal currents driven by sealevel fluctuations and by tilting . Experiments were run in a 20 by 3 m straight flume, the Metronome, for a range of tilting periods and one or two boundaries open at constant head with free inflow and outflow. Also experiments were run with flow driven by periodic sealevel fluctuations. We recorded surface flow velocity along the flume with Particle Imaging Velocimetry and measured water levels along the flume. We compared the results to a one-dimensional model with shallow flow equations for a rough bed, which was tested on the experiments and applied to a range of length scales bridging small experiments and large estuaries. We found that Reynolds' method results in negligible flows along the flume except for the first few meters, whereas flume tilting results in nearly uniform, reversing flow velocities along the entire flume that are strong enough to move sand. Furthermore, tidal excursion length relative to basin length and the dominance of friction over inertia is similar in tidal experiments and reality. The sediment mobility converges between Reynolds' method and tilting for flumes of hundreds of meters long, which is impractical. Smaller flumes of a few meters length on the other hand are much more dominated by friction than natural systems, meaning that sediment suspension would be impossible in the resulting laminar flow on tidal flats. Where Reynolds' method is limited by small sediment mobility and high tidal range relative to water depth, the tilting method allows independent control over the variables flow depth, velocity, sediment mobility, tidal period and excursion length, and tidal asymmetry. A periodically tilting flume thus opens up the possibility of systematic biogeomorphological experimentation with self-formed estuaries.
\end{abstract}

%novelty, originality and importance: This paper reports on a new experimental facility that allows live-bed landscape experiments of tidal systems. Although some experiments in a much smaller pilot setup are published, this much larger facility is presently used for experiments similar to the breakthrough braided and meandering river experiments in several labs, only now with tidal flow. This paper provides experimental and numerical evidence that the setup by Osborne Reynolds, used for over a century, is flawed whilst the novel setup works to create flows with mobile sediment. Simulations with a bespoke numerical model show that upscaling from experiment to natural systems requires vertical distortion, while characteristic relative length scales of tidal wavelength and tidal excursion length remain reasonably close to those of natural systems. Moreover, the tilting allows independent control of these variables, sediment mobility and tidal asymmetry.

\introduction  %% \introduction[modified heading if necessary]

[revised manuscript text omitted]

%\label{fig:modelwaveasym}
%\end{figure}

%\begin{figure}[t]
%\includegraphics[width=12cm]{modeltidalanalysis.png}
%\caption{Tidal component analysis for increasing tilting slope amplitude with 30~s period. (a,c,e) Velocity amplitude of the T1 (30~s), T2 and T3 components. Note different amplitude scale in (a). (b,d,f) Phases of the same components.}
%\label{fig:modeltidalanalysis}
%\end{figure}

\clearpage

\begin{figure*}[t]
\includegraphics[width=18cm]{unevenfloor.png}
\caption{Sensitivity of flow to depth variation in the basin (drawn lines) compared with the ideal depth (dashed lines) for 30~s period model runs. Inset in (c) shows modified bed elevation. Runs with original depth are same as in Fig.~\ref{fig:closedtilt100mm30s}. (a,c) Flow velocity at the water surface along the flume for selected phases of the tidal cycle. Legend as in Fig.~\ref{fig:closedtilt100mm30s}. (b,d) Water level and flow velocity in one tidal cycle showing hysteresis. %Legend as in Fig.~\ref{fig:hysteresis}.
(a,b) Tilting at 0.09~m amplitude with upstream boundary closed. (c,d) Reynolds method with sealevel fluctuation at 0.01~m amplitude with the landward boundary closed. }
\label{fig:unevenfloor}
\end{figure*}

\begin{figure*}[t]
\includegraphics[width=18cm]{sedimentmobility.png}
\caption{Sediment mobility calculated for tidal flows driven by periodic tilting (drawn lines) and by the Reynolds method with periodic sea surface fluctuation (dashed lines). Legend as in Fig.~\ref{fig:closedtilt100mm30s}. Gray area indicates immobile sediment.}
\label{fig:sedimentmobility}
\end{figure*}

\clearpage

\begin{figure*}[t]
\includegraphics[width=15cm]{scalingflumeNEW.png}
\caption{Scaling from Metronome to natural estuaries by the numerical model as $n_h=n_T=n_L$ for all settings, a constant $a$ for the tilting setup (dashed lines for tilt1 and dotted lines for tilt2) and a constant $a/h$ for the Reynolds method (drawn lines). Legend indicates length of the basin in m and relative downstream distance is from 0:landward to 1:inlet.}
\label{fig:scalingnew}
\end{figure*}

\begin{figure*}[t]
\includegraphics[width=14cm]{floodasymtime.png}
\caption{Flood-dominance accomplished in the tilting flume by adding an overtide to the tilting with an amplitude of 20\% of the principal tide with a phase delay of $\pi/2$. Ebb flow is positive. The transport is cumulative in time to show the flood-dominance in two tidal cycles.}
\label{fig:asymmetrictilt}
\end{figure*}

\end{document}